# Improving multimodal datasets with image captioning

**Thao Nguyen**
University of Washington
thaottn@cs.washington.edu

**Samir Yitzhak Gadre**
Columbia University
sy@cs.columbia.edu

**Gabriel Ilharco**
University of Washington
gamaga@cs.washington.edu

**Sewoong Oh**
University of Washington,
Google Research
sewoong@cs.washington.edu

**Ludwig Schmidt**
University of Washington,
Allen Institute for Artificial Intelligence
schmidt@cs.washington.edu

## Abstract

Massive web datasets play a key role in the success of large vision-language models like CLIP and Flamingo. However, the raw web data is noisy, and existing filtering methods to reduce noise often come at the expense of data diversity. Our work focuses on caption quality as one major source of noise, and studies how generated captions can increase the utility of web-scraped datapoints with nondescript text. Through exploring different mixing strategies for raw and generated captions, we outperform the best filtering method proposed by the DataComp benchmark by 2% on ImageNet and 4% on average across 38 tasks, given a candidate pool of 128M image-text pairs. Our best approach is also $2\times$ better at Flickr and MS-COCO retrieval. We then analyze what makes synthetic captions an effective source of text supervision. In experimenting with different image captioning models, we also demonstrate that the performance of a model on standard image captioning benchmarks (e.g., NoCaps CIDEr) is not a reliable indicator of the utility of the captions it generates for multimodal training. Finally, our experiments with using generated captions at DataComp's `large` scale (1.28B image-text pairs) offer insights into the limitations of synthetic text, as well as the importance of image curation with increasing training data quantity. The synthetic captions used in our experiments are now available on HuggingFace[1].

## 1 Introduction

Pre-training large multimodal networks on image-text pairs sourced from the web has become a standard approach to obtaining high performance on vision tasks [3, 37, 24, 40]. However, raw web data can be noisy or uninformative (Figure 1). Most existing data preprocessing efforts revolve around human-defined heuristics based on image and text content separately—e.g., caption length, presence of nouns, sentence complexity, image aspect ratio, minimum image size [8, 46, 10, 47]—or the reliability of the data source [14]. More complex filtering approaches target poorly aligned image-text pairs, by using OpenAI's CLIP models [40] to rank the cosine similarity score between image and text embeddings [46], or ensuring mentions of image objects in the captions [47]. These approaches discard between 60% to 90% of the initial data collected, regardless of whether the images themselves are perfectly suitable for training.

In this work, we seek to restore the utility of such discarded examples with the help of synthetic captions, and explore the impact on performance of expanding the training set this way. To do so, we leverage the DataComp benchmark [18], where initial data processing is kept to a minimum

---

[1]https://huggingface.co/datasets/thaottn/DataComp_medium_pool_BLIP2_captions,
https://huggingface.co/datasets/thaottn/DataComp_large_pool_BLIP2_captions

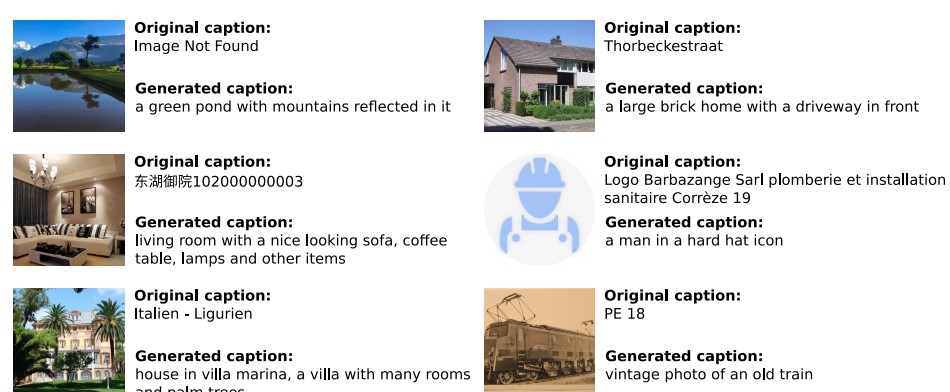

Figure 1: **Raw captions crawled from the web contain significant noise; cosine similarity filtering helps reduce noise but discards many images that are useful for training.** Here we show some images that would be filtered out if only the top 30% examples from the candidate pool with highest image-text cosine similarities are used for training. In these pairs, captions generated by BLIP2 tend to be more faithful to the respective images compared to raw captions obtained from the Internet. In Appendix A, we show 20 other samples drawn completely at random from the discarded pool.

(i.e. only filtering out NSFW examples and train-test overlap). This allows us to perform controlled experiments on the raw Common Crawl data directly and bypass subjective human-design choices that may be employed in the creation of other datasets (e.g., see LAION-5B [46]). We study several image captioning models and find that recent releases (e.g., BLIP2 [30] and OpenCLIP-CoCa [38]) can generate synthetic captions that improve CLIP training, and lead to a significant boost in zero-shot performance over existing data curation methods. In particular, at the `medium` scale (128M samples seen), training on the *entire candidate pool* with synthetic captions is sufficient to outperform common filtering baselines that were done on raw data (e.g., selecting top 30% examples with highest image-text similarity based on OpenAI's CLIP-ViT/L14). Section 5 describes our experiments with a variety of mixing strategies to combine signals from both raw and synthetic text.

To explain the performance benefits of synthetic captions, we measure caption noise and diversity in various training sets, and demonstrate the significance of both factors in achieving good performance. While existing data filtering methods are effective at reducing noise, they also hurt the diversity of the original training data in the process (e.g., by reducing concept coverage). Synthetic captions help alleviate this drop in diversity by increasing the number of useful captions available for training. In section 6, we analyze various properties of caption data, as well as specific advantages of training with synthetic captions (e.g., improved retrieval capabilities).

Remarkably, our empirical investigation in Section 4 shows that choosing a captioning model to yield competitive downstream performance is non-trivial, as better performance on image captioning benchmarks does not necessarily mean better caption quality for CLIP training. We also note that while this work focuses on the the quality of captions used in multimodal training, image quality is another equally important topic of study. As the size of the data pool we experiment with grows exponentially, we start to observe changes in the relative importance of text quality versus image quality in building a good pre-training dataset. We will comment on this in Section 7.

To summarize, our findings serve as a first step towards improving the quality of *web-scale* datasets via the use of synthetic captions. In the process, we offer insights on several research directions:

- *What are the considerations for choosing a captioning model?* We find that specializing a pre-trained network towards image captioning via fine-tuning, and optimizing for high CIDEr score on standard benchmarks in general, end up producing captions that are less effective for multimodal training. Reference-free captioning metrics (e.g., CLIP-S [21]) more reliably reflect the training quality of the generated captions.

- *How to combine signals from multiple sources of captions?* We investigate different strategies for filtering and mixing raw and synthetic captions. This leads to performance gains on DataComp benchmark at `small` (12.8M pool size), `medium` (128M pool size) and `large` (1.28B pool size) scales, compared to existing approaches that utilize only raw data. On ImageNet, the performance benefits diminish with scale. On retrieval tasks, however, the gains are significant across all scales.

- *What makes synthetic captions effective?* Our analysis of text properties shows that on an individual level, synthetic captions are less noisy and contain more visual information. However, at the population level, synthetic captions are less diverse than raw captions. Consequently, using *both* sources of captions helps improve the overall caption quality, measured in terms of text diversity as well as image-text alignment.
- *How do benefits of synthetic captions scale?* Unlike what was found in the original DataComp experiments, given access to generated captions, the best filtering approach differs across scales. Experimenting with data quantities ranging from 12.8M to 1.28B also allows us to observe some limitations of synthetic captions. We posit that image-based filtering, as well as the diversity gap between model-generated and web-scraped captions, play an increasingly important role in large data regimes.

More broadly, our results have important implications for future work as additional progress (captured by the right metric) in image captioning can further enhance the quality of text used for vision-language pre-training. Moreover, the effectiveness of synthetic captions unlocks another massive source of training data: uncaptioned web images from Common Crawl. This can ultimately empower more large-scale multimodal training by improving the availability of properly aligned and sufficiently diverse image-text data.

## 2   Related work

**Synthetic data.**   Previous work has explored using synthetic data to create new datasets or augment existing ones [15, 41, 36, 25, 56, 19, 12, *inter alia*]. Closer to our work, He et al. [20], Azizi et al. [5], Bansal and Grover [6] use image generation models to create synthetic images for classification tasks. In the context of CLIP, Santurkar et al. [44] show that a model trained on synthetic captions can outperform a model trained on human-provided captions. The captions were generated procedurally for the 120K images in the MS-COCO training set [11], using multi-object image labels verified by Mechanical Turk workers, which would be difficult to obtain for web-scale datasets like LAION-5B [46] or CommonPool [18] that are about four orders of magnitude larger. Most similar to our work is the LAION-COCO dataset [45], containing 600M image-text pairs from LAION-5B [46] with synthetic captions generated using BLIP [29] and ranked using CLIP models [40, 23]. While [45] heavily filters the raw data pool before generating captions, we work with uncurated web datasets.

**Image captioning.**   Building models able to generate captions from images has been a long-standing subject of research [28, 27, 31, 13, 53, 52, *inter alia*]. More recently, models like BLIP2 [29, 30], Flamingo [3], and CoCa [55, 38] have made significant progress on this task. Zhu et al. [57] couple large language models with image captioning models to generate more enriched image descriptions. Our work builds on existing image captioning systems to generate synthetic captions for web-crawled training images.

**Improving image-text datasets.**   Given the importance of the pre-training data for multimodal networks [33, 17, 18], several authors have proposed techniques for improving the quality of image-text datasets. [9] filter out samples that contain text regions in the image, and advocate for the benefits of increasing the number of samples given a fixed compute budget. Abbas et al. [1] identify and remove samples that are semantically similar to each other. [39] propose a filtering technique called Complexity, Action, and Text-spotting (CAT), designed to select only informative image-text pairs. Many image-text datasets also have their own preprocessing techniques, often not fully disclosed [40, 24, 37, 10, 14, 46]. All of these filtering approaches are complementary to the use of synthetic captions proposed by this work. Liu et al. [32] introduce TaiSu, a Chinese image-text dataset where where an image captioning model is used to generate a supplementary description for the images, which are subsequently translated into Chinese with machine translation. Concurrent to our work, Fan et al. [16] present a form of data augmentation for training CLIP models where the captions are rewritten by a large language model. However, the rewriting process assumes access to some raw text and is not conditioned on the images, which may limit its effectiveness when the original captions are not descriptive (e.g., see Figure 1). In contrast, our work uses image captioning models, which are able to generate relevant captions for images regardless of the original text associated with them. We also work with raw Common Crawl data instead of preprocessed datasets, to study the trade-offs between raw and generated captions in a systematic manner. Finally, Gadre et al. [18] introduces DataComp, a benchmark for designing better pre-training datasets for CLIP, which we use in experiments throughout the paper.

# 3 Experiment setup

**Data.** Most of our experiments involve the CommonPool provided by the DataComp benchmark [18]. The data contains image-text pairs sourced from Common Crawl dumps between 2014 and 2022, deduplicated and randomly shuffled. The `small`, `medium` and `large` scales of the benchmark contain 12.8M, 128M and 1.28B candidate pairs respectively. Data preprocessing is kept to a minimum, involving only NSFW filtering, evaluation set deduplication and face blurring, to allow maximum flexibility for dataset design. We also experiment with LAION-COCO [45] and discuss in Appendix G why it is not ideal for studying how to improve the quality of raw training data.

**Captioning models.** We experiment with BLIP and BLIP2 [30] using HuggingFace's Transformers framework. Both models were pre-trained on 129M image-text pairs from the web including MS-COCO [11] and LAION-400M [46], in addition to the bootstrapped version of the web data with captions generated by BLIP. We also look at OpenCLIP-CoCa [38, 23], which was trained on 13B samples seen from LAION-2B [46]. For each architecture, we experiment with both the pre-trained model and the one that has been fine-tuned on MS-COCO. Caption generation uses top-K sampling with K = 50, minimum caption length 5 and maximum caption length 40.

**Training.** Given CommonPool data of a particular scale, we generate synthetic captions for the images in the pool using the captioning models described above. Then we train a CLIP model on the resulting image-text datasets, using ViT-B/32 as the image encoder for the `small` and `medium` scales, and ViT-B/16 for the `large` scale. Following DataComp's setup [18], the compute budget, architecture and hyperparameters for each scale are fixed in order to isolate data quality as the main factor influencing performance. Given a candidate pool of $N$ image-text pairs, the CLIP model is then trained with $N$ samples seen in total. Refer to Appendix B for more details.

**Evaluation.** We adopt DataComp's zero-shot evaluation suite, and report both ImageNet accuracy and the average accuracy over 38 classification and retrieval tasks proposed by the benchmark [18]. We also pay particular attention to retrieval performance on Flickr30K [54] and MS-COCO [11]. The retrieval score reported is the average of text-to-image Recall@1 and image-to-text Recall@1.

Unless specified otherwise, in the subsequent sections, "CLIP score filtering" or "top x%" refers to selecting top x% examples from the initial training set, based on the cosine similarity between image and text embeddings output by OpenAI's CLIP ViT-L/14 model [40], and "BLIP2" refers to captions generated by BLIP2, using top-K sampling with softmax temperature = 0.75, which we have found to yield the best downstream performance compared to other sampling temperatures (see Appendix C).

# 4 Impact of model specialization on captions generated for CLIP training

| Captioning model | NoCaps CIDEr [51] | CLIP-S [21] | Cosine similarity | No. of unique trigrams | ImageNet accuracy | Flickr retrieval |
|---|---|---|---|---|---|---|
| BLIP, ViT-L/16 (finetuned) | 113.2* | 0.698 | 0.231 | $2.82 \times 10^6$ | 0.207 | 0.498 |
| BLIP2, ViT-g | 80.6 | 0.737 | 0.251 | $2.72 \times 10^6$ | 0.281 | 0.507 |
| BLIP2, ViT-g (finetuned) | 119.7* | 0.711 | 0.235 | $1.97 \times 10^6$ | 0.227 | 0.549 |
| OpenCLIP-CoCa, ViT-L/14 | 0.354* | 0.752 | 0.260 | $4.45 \times 10^6$ | 0.321 | 0.395 |
| OpenCLIP-CoCa, ViT-L/14 (finetuned) | 106.5* | 0.702 | 0.232 | $1.81 \times 10^6$ | 0.252 | 0.542 |

Table 1: **CIDEr score does not reliably predict how effective a captioning model is at generating synthetic captions for multimodal pre-training; fine-tuning image captioning models leads to lower ImageNet accuracy when training CLIP on the generated captions.** * indicates numbers obtained from the corresponding previous work and from contacting the authors. We fix the architecture and compare captioning models with and without fine-tuning on MS-COCO [11], as sources of text supervision for CLIP. Fine-tuning pre-trained networks on the task of image captioning ends up producing synthetic captions that are worse for training CLIP (see resulting ImageNet accuracy), possibly due to lower text diversity. On the contrary, retrieval performance is higher when using captions generated by fine-tuned models.

Given the abundance of image captioning models to choose from, a natural question to ask is: does performance on standard image captioning benchmarks correlate with how useful the generated captions are as text supervision for CLIP training?

In particular, CIDEr [51], together with other reference-based metrics like SPICE [4] and BLEU-4 [35], has been widely adopted as a yardstick for determining state-of-the-art on image captioning benchmarks [55, 3, 29, 30, 22]. Consequently, previous work [55, 29, 30] also experiment with fine-tuning captioning models on MS-COCO and obtain competitive CIDEr scores on common evaluation sets like NoCaps [2].

We compare the utility of synthetic captions produced by BLIP2 and OpenCLIP-CoCa with and without fine-tuning on MS-COCO, by training CLIP on the generated captions and evaluating the trained model on ImageNet classification and Flickr retrieval (Table 1). Fine-tuned captioning models produce captions that boost the retrieval capabilities of CLIP, but hurts its ImageNet classification performance. We hypothesize that fine-tuning on MS-COCO reduces the diversity of the generated text, as evidenced by the lower number of unique trigrams across 1M caption samples (Table 1). Notably, captioning models that are not fine-tuned have very poor CIDEr scores; going with this metric would have suggested that these models are not suitable for caption generation at all.

While many image captioning metrics like CIDEr, SPICE and BLEU-4 emphasize similarity between generated captions and reference captions provided by humans, prior work has also proposed reference-free metrics—for example, CLIP-S [21], which uses a trained CLIP model to assess the compatibility between an image and the generated caption. We compute CLIP-S for the `medium` candidate pool with different synthetic captions, and find that it is more reflective of the ImageNet performance trend. Fine-tuned captioning models produce captions that have lower CLIP-S and image-text cosine similarity in general.

Since BLIP2 (no fine-tuning) produces sufficiently good text supervision for CLIP to do well on both ImageNet and Flickr, we use it as the captioning model of choice in subsequent experiments that look at how to combine raw and synthetic captions.

## 5  Filtering raw and synthetic captions

Here we explore in more detail different ways of filtering and combining raw and generated captions, at the `medium` scale of DataComp [18]:

- *No filtering:* we train on the entire, unmodified pool (i.e., 128M samples).
- *CLIP score filtering:* we select the top x% of examples with highest image-text cosine similarity.
- *CLIP score intersect with ImageNet1k clustering:* Gadre et al. [18] propose clustering image embeddings and only selecting images whose cluster center is a nearest neighbor to one image from ImageNet1k. The authors then find the intersection between this set of images and those that are in the top x% based on CLIP score. This is the best baseline using raw captions on DataComp.
- *Combining raw and synthetic captions:* we use raw captions for the top x% of examples based on CLIP score. For the remaining images (that would otherwise be filtered out), we generate corresponding BLIP2 captions and add them back to the training pool. We also experiment with filtering these additional image-text pairs with the same cosine similarity threshold set in the first step (i.e., BLIP2 (X%, FILTERED) in Figure 2).

In Appendix D, we describe other baselines we have tried and report how well each approach does with varying cosine similarity thresholds. Figure 2 (left) shows the relative performance of select baselines (the degree of CLIP score filtering has been tuned and only the best accuracy is plotted). We find that the best performance at `medium` scale, measured by either ImageNet or average accuracy, is achieved by mixing raw and synthetic captions, subject to a cosine similarity threshold. Appendix Figure 9 shows the results for Flickr and MS-COCO retrieval, where including BLIP2 captions in the training pool also offers significant performance benefits.

In the right plot of Figure 2, we compare ImageNet performance at various filtering thresholds for methods that involve only one source of captions and those that involve both. We observe that given image-raw-text pairs filtered with certain cosine similarity threshold (blue line), adding BLIP2 captions for some (red line) or all of the remaining images (green line) always helps. It is worth noting that as we lower the threshold and include more raw captions in the training mix, the performance starts to become lower than using just synthetic captions (orange line). Overall we find that filtering is still a necessary step even in the presence of synthetic captions that are supposedly more relevant to the training images.

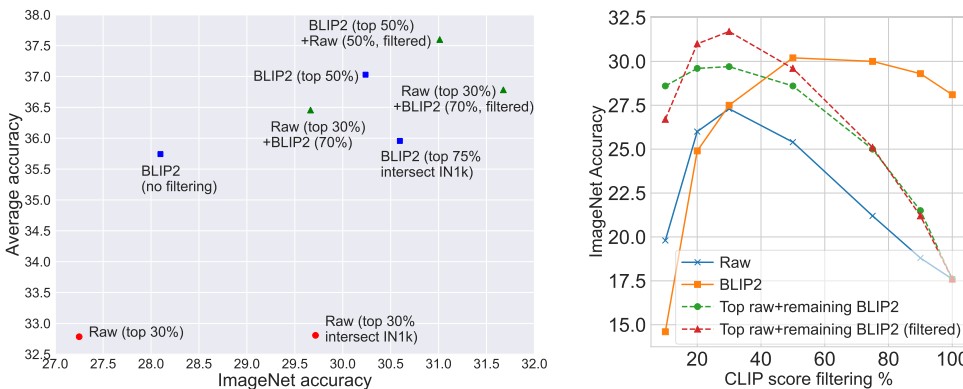

Figure 2: **At the 128M scale of DataComp, we obtain improvement on ImageNet and average accuracies compared to the best filtering method on raw data, by using a mixture of raw and synthetic captions, selecting only image-text pairs with cosine similarity above a certain threshold.** (Left) We visualize how various data filtering strategies perform at `medium` scale, on ImageNet and across 38 tasks. Including BLIP2 captions in the training data significantly outperforms competitive baselines from DataComp trained on only raw text [18]. (Right) As we vary the percentage of top examples chosen from the pool (based on CLIP score), we see consistent benefits from ($i$) using BLIP2 captions for samples that would be discarded otherwise, ($ii$) applying the same filtering threshold to new image-text pairs containing BLIP2 captions to keep noise level low. The exact accuracy numbers could be found in Appendix D.

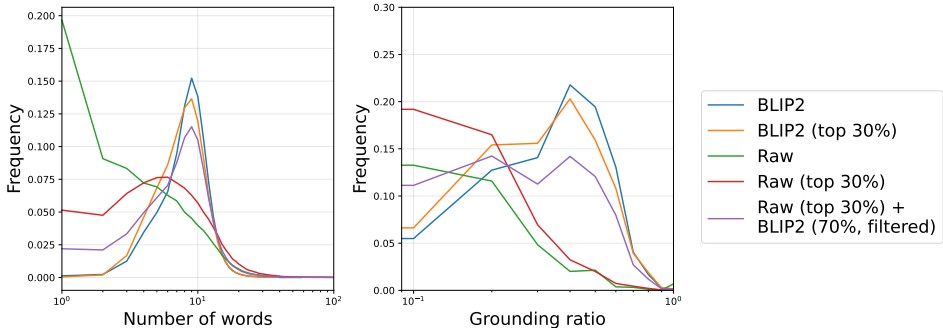

Figure 3: **Individual synthetic captions can contain more information (especially visual one) compared to raw captions.** We calculate the number of words and the fraction of those being visual tokens in each caption for different training sets. Individual BLIP2 captions tend to yield higher numbers on these two metrics compared to individual web-crawled captions, suggesting that on a caption-per-caption basis, synthetic data may contain richer information.

# 6 What makes synthetic captions effective?

## 6.1 Defining caption quality

As seen from sample images in Figure 1, web-scraped text may not contain specific visual information (e.g., "Italien - Ligurien") or may not reflect the content of the image (e.g., "Image Not Found"). We seek to understand how generated captions can help overcome these problems.

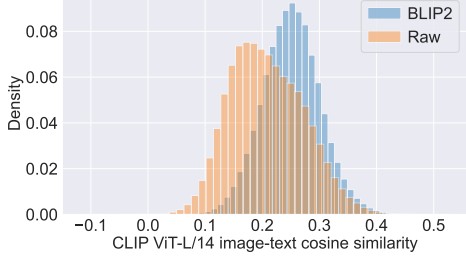

Figure 4: **Generated captions overall exhibit higher image-text alignment than raw captions; this indicates that the former may be less noisy as a training source.** We randomly sample 1% of the 128M candidate pool and given the same set of images, compare the cosine similarity distribution between raw caption data and BLIP2 caption data. We find that overall BLIP2 captions have much higher image-text cosine similarity (mean similarity 0.251 vs 0.208).

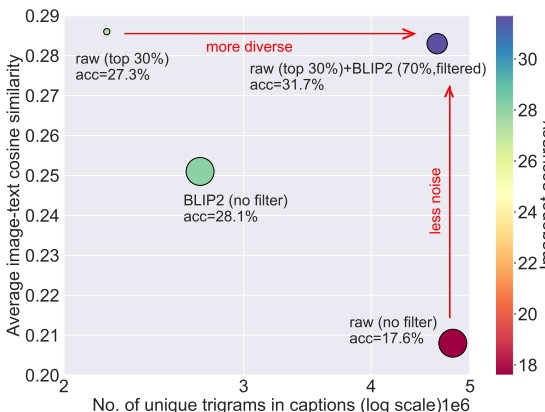

Figure 5: **Combining raw and synthetic captions subject to a cosine similarity threshold helps reduce noise level while boosting data diversity, both of which are essential for achieving good performance.** In this plot, circle size denotes the relative size of the resulting training set. While removing noisy image-text pairs, CLIP score filtering also lowers the diversity of the caption set substantially, as measured by the number of unique trigrams in the pool. Adding more useful training data by using BLIP2 captions for filtered out images, while respecting the existing CLIP score threshold, helps overcome this limitation and improves the training data quality along both axes.

To approximate the richness of information conveyed in the text data, for each sample in a 1M random subset, we measure the number of words, as well as the grounding ratio [49] (i.e., fraction of tokens that describe visual concepts, with the vocabulary defined by MS-COCO), in the corresponding captions. In Figure 3, we observe that synthetic captions and raw captions follow vastly different distributions, with the former generally containing more words (left pane) and more visual tokens (right pane) per sample. Performing CLIP score filtering on raw captions leads to improvements on both of these properties; so does mixing raw and synthetic captions. Regarding the issue of poor image-text alignment, we approximate the alignment using cosine similarity between image and text embeddings from CLIP, and find that web-crawled captions indeed have lower similarities overall compared to model-generated ones (Figure 4).

The analyses above measure properties of individual captions. We next aim to capture a single diversity metric over *all* text in the training set. We again select a random subset, the size of which scales with the training set size, and calculate the number of unique trigrams across all captions in the subset. With this diversity metric, we find that BLIP2 captions actually lag behind raw captions (Figure 5). Using only the top 30% raw captions (based on CLIP score) is even more detrimental.

We summarize these different aspects of caption quality in a noise versus diversity framework (Figure 5), which also offers some intuition for our best baseline uncovered in Section 5. CLIP score filtering that has been widely adopted in prior work [46, 18] is effective at improving performance on raw data by removing noisy examples (i.e., those with poor image-text alignment). However, this procedure also lowers diversity (note: Figure 5 only provides a measure of text diversity, but image diversity is affected as well). By generating synthetic captions for the images that would be discarded otherwise, and subsequently only using pairs where the cosine similarities still meet the threshold, we manage to keep the overall noise level similarly low, while adding more diversity to the training pool. Progress along both axes enables further performance improvement compared to just filtering raw data.

## 6.2 Performance analysis

After diving deeper into properties of synthetic captions, we next analyze the training implications of these captions in more detail. We examine two models, one trained using only raw captions and the other using only BLIP2 captions, with both training sets having been filtered with CLIP score for top 30% pairs, and achieving similar performance on ImageNet (27.3% vs 27.5%). Averaged across 38 evaluation tasks, training on generated captions sees a 2.8% improvement. We break down performance difference between the two models on individual tasks (Figure 6), and observe that BLIP2 captions also perform better on ImageNet-derived distribution shifts and text recognition (e.g., MNIST, SVHN). Notably, among the tasks with the biggest performance gains are Flickr and MS-COCO retrieval. We provide a similar analysis in Appendix Figure 10, where expanding a filtered training set with additional images and their BLIP2 captions improves performance on 30/38 tasks.

The two models compared above share similar ImageNet accuracy but may not be trained on the same images. In Figure 7, we fix the set of training samples to be the top 30% with highest cosine similarity between image and *raw* text. Replacing the raw captions with BLIP2 captions increases retrieval performance on Flickr and MS-COCO by more than 1.5× (first two columns of each task). We include retrieval performance of training on the entire pool with BLIP2 captions (generated using

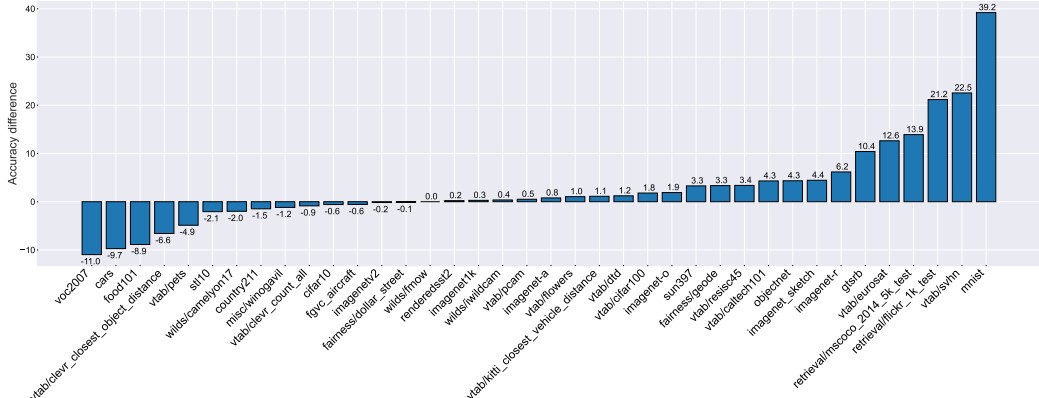

Figure 6: **Given similar ImageNet accuracy, training with synthetic captions improves performance on 23 out of 38 tasks compared to training with raw captions, including ImageNet distribution shifts, text recognition and retrieval tasks.** We compare performance on each task between training with only BLIP2 captions and training with only raw captions; both datasets have been filtered with CLIP score to select the top 30% examples. Even though the two training sets yield similar ImageNet accuracy (∼27%), using generated captions leads to 2.8% improvement on average accuracy, including minor gains on ImageNet distribution shifts and significant gains on MNIST, SVHN, Flickr and MS-COCO retrieval.

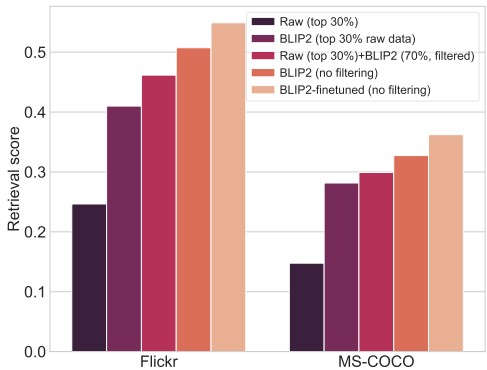

Figure 7: **Synthetic captions display a clear advantage over raw captions on retrieval tasks.** We highlight the superior performance on Flickr and MS-COCO retrieval obtained from training on captions generated by BLIP2 (pre-trained model or model that has been fine-tuned on MS-COCO), compared to training on raw captions. In particular, the first two columns of each task represent two models trained on the same set of images (i.e., those whose cosine similarity between image and *raw* text embeddings are in the top 30%), just with different captions. This suggests that substantial gains on retrieval tasks can be obtained solely by using better aligned captions.

either the pre-trained or the fine-tuned captioning model), as well as that of training on a mixture of raw and BLIP2 captions, to demonstrate the consistent gains that synthetic captions have to offer.

## 7  Performance at scale

We next apply select baselines described in Section 5 to a wide range of candidate pool sizes, ranging from 12.8M to 1.28B samples. In particular, we examine training on the entire pool with only raw captions or only BLIP2 captions, CLIP score filtering, using the intersection of top CLIP score examples and examples that lie in clusters close to the ImageNet train set, as well as mixing raw and synthetic captions—our best baseline from the `medium` scale. The filtering percentage for each method is tuned on the `medium` scale candidate pool and then applied to experiments at other scales. Given a starting pool of $N$ samples, we limit the training budget to $N$ steps. Note that the 400M and 1.28B scales use the `large` training settings from DataComp (see [18]).

We focus on ImageNet classification and Flickr retrieval performance (note: MS-COCO training set was included in BLIP2's pre-training data so we have excluded MS-COCO retrieval from this comparison). At larger data quantity regimes, using synthetic captions continues to substantially outperform existing raw-text filtering baselines at retrieval (Figure 8, right plot). On ImageNet, however, adding BLIP2 captions to the training mix sees diminishing returns: RAW (TOP 30% INTERSECT IN1K) + BLIP2 (REMAINING 70%, FILTERED, INTERSECT IN1K) outperforms existing state-of-the-art baseline trained on raw data, RAW (TOP 30% INTERSECT IN1K), by 2.5% at 400M scale and 1.2% at 1.28B scale (Figure 8, left plot).

To give some intuition for this result, we offer two candidate hypotheses:

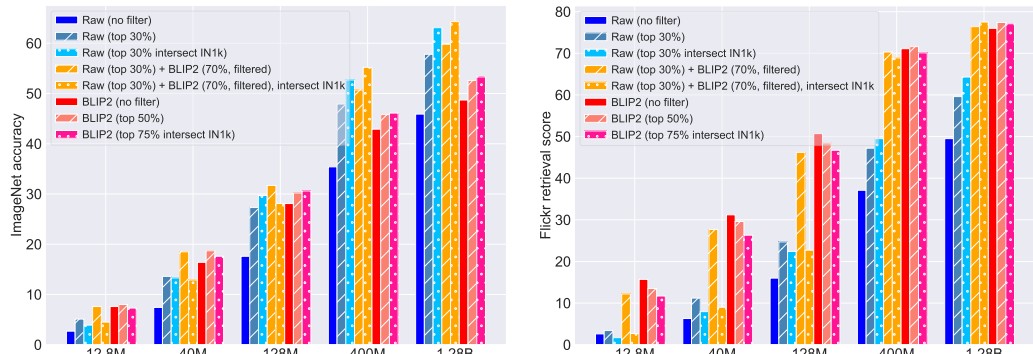

Figure 8: **With access to synthetic captions, we find that the best data filtering method for ImageNet classification varies with the scale of the candidate pool; when it comes to retrieval, however, using synthetic captions is highly beneficial across all scales.** We apply select baselines from Section 5 to a range of candidate pool sizes, and find that the best method on Flickr retrieval always involves synthetic captions (right plot). On ImageNet (left plot), selecting meaningful images (e.g., those that lie close to the ImageNet train set in the embedding space) becomes increasingly important at larger scales (see dotted versus striked columns). As the data pool size increases, using BLIP2 captions seems to yield diminishing returns, possibly due to the saturation of text diversity obtained from image captioning models.

- As noted in Section 6, both noise level (i.e., image-text alignment) and text diversity are important for performance. Noise level stays about the same across all scales. In contrast, the diversity gap between model-generated text and web-scraped text may become more significant with increasing data quantities. We repeat the caption quality analyses from Section 6 with varying random subset size, and find that when using the number of unique nouns and unique trigrams as proxies for text diversity, synthetic captions exhibit a worse scaling trend than raw captions (Appendix Figure 12).
- Image quality becomes increasingly important at larger scales:
  (i) from 12.8M to 128M scale, training on the *entire candidate pool* with BLIP2 captions outperforms competitive filtering baselines done on raw data (e.g., RAW (TOP 30%)). This is not the case for larger scales.
  (ii) starting from 128M scale, baselines that also curate image content (i.e., intersection of top CLIP score examples and those that lie in clusters close to the ImageNet train set) consistently outperform baselines that involve only CLIP score filtering, using either raw captions or BLIP2 captions.

Exact performance numbers can be found in Appendix F and Table 3. Overall, we find that given a fixed training budget, making more datapoints useful by carefully replacing noisy raw captions with synthetic captions—i.e., RAW (TOP 30%) + BLIP2 (70%, FILTERED) versus RAW (TOP 30%)— still offers significant classification and retrieval performance gains across *all* scales. However, for synthetic captions to perform competitively on ImageNet at larger data regimes, we need to start paying attention to image content, as well as enhancing the diversity of the generated text.

## 8 Conclusion

In this work, we demonstrate the effectiveness of synthetic captions in improving caption quality for multimodal training, as well as certain capabilities of the resulting model (e.g., retrieval). Notably, we find that fine-tuning general purpose models towards the task of image captioning actually makes them less effective at producing good captions for CLIP training. Our experiments with various candidate pool sizes, ranging from 12.8M to 1.28B image-text pairs, show that including generated captions in the training data can be highly effective at `small` and `medium` scales. However, with larger data quantities, the diversity gap between model-generated and web-scraped text begin to hinder performance gains, and it becomes increasingly harder to obtain state-of-the-art ImageNet accuracy by just improving text supervision alone.

**Limitations.** Our experiments do not involve an exhaustive list of image captioning systems currently available. Given a captioning model of sufficient capability—i.e., it can generate captions for training CLIP to reach a good performance—a major theme of our work is understanding how

to combine signals from both raw and synthetic captions, as well as the differences between these two sources of text. We note that even with improved caption quality, multimodal web datasets may still contain harmful stereotypes, some of which have been extensively discussed in prior work [7]. In Appendix H, we conduct some preliminary investigation on the change in race and gender bias between training on synthetic captions and training on only web-crawled text. Besides, generated captions can introduce new biases that they inherit from the captioning models, and using these captions to train the next generation of models can amplify the biases. The risks from using model outputs to replace human annotations have been studied in a simplified settings in [50, 48].

**Future work.** Our findings motivate a number of interesting future directions. One concrete question is whether combining synthetic caption data from multiple image captioning systems can enhance text diversity. Another direction is proposing new algorithms to combine information from raw and generated captions, beyond what we already studied in Section 5 and Appendix D. Future work could also explore using text-to-image generation [42, 34, 43] to create synthetic training images for concepts that are underrepresented in existing captions, in order to boost data diversity and close knowledge gaps in the downstream model.

## Acknowledgments and Disclosure of Funding

We thank Stability AI for the generous assistance with compute resources. We are grateful to Josh Gardner and Simon Kornblith for providing feedback on the manuscript. We also thank Maciej Kilian, Anas Awadalla, Alex Fang, Dhruba Ghosh and Jonathan Hayase for helpful discussions while working on this paper. SYG is supported by a NSF Graduate Research Fellowship. This work is supported in part by Open Philanthropy, the Allen Institute for AI, and NSF grants DMS-2134012 and CCF-2019844 as a part of NSF Institute for Foundations of Machine Learning (IFML).

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

# A    More examples of image-text pairs (no cherry picking)

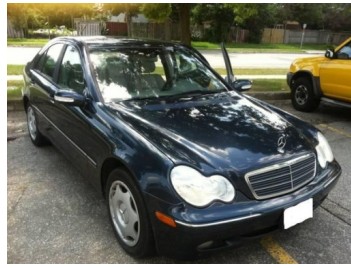

Raw: *2003 Mercedes-Benz C240 sedan, Leather, MUST BE SEEN - $6199*
BLIP (finetuned): *a couple of cars parked in a parking lot with trees and cars*
BLIP2: *2002 mercedes-benz c-class for sale*
BLIP2 (finetuned): *a blue mercedes benz car parked in a parking lot next to yellow cars*
OpenCLIP-CoCa: *find used 2 0 0 1 mercedes benz c 2 4 0 base sedan 4 door 2 5 l for 2 0 0 1 mercedes benz c 2*
OpenCLIP-CoCa (finetuned): *a blue mercedes parked on the side of a road .*

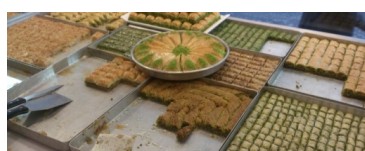

Raw: *Gaziburma Ünal is one of Gespeicherte Orte von Can.*
BLIP (finetuned): *dozens of trays of different types of treats at a food stand*
BLIP2: *some trays of pastries and a knife*
BLIP2 (finetuned): *there are many trays filled with food items from the store*
OpenCLIP-CoCa: *baklava , sweets , pastries*
OpenCLIP-CoCa (finetuned): *there are trays full of different types of food .*

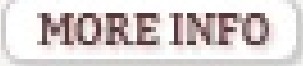

Raw: *Open Virgin of Kazan, Wooden Egg with Stand, Blue*
BLIP (finetuned): *a gray and white logo with the words more info in a rectangular shape*
BLIP2: *a white button with the word more info*
BLIP2 (finetuned): *more information is shown on a white button with an orange background*
OpenCLIP-CoCa: *home - page - button . png*
OpenCLIP-CoCa (finetuned): *a picture of a close up of a text message*

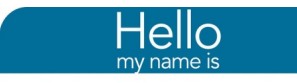

Raw: *2016.07.01 Nametags with Pronouns - Avery 5392_non-branded*
BLIP (finetuned): *there are no pictures here to provide a caption for*
BLIP2: *hello, my name is name, my pronouns are pronouns*
BLIP2 (finetuned): *a blue and white label with a blue and white text*
OpenCLIP-CoCa: *1 5 + hello my name is names pronunciations and meanings*
OpenCLIP-CoCa (finetuned): *hello my name is , my pronouns are .*

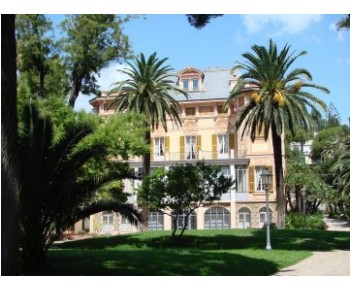

Raw: *Italien - Ligurien*
BLIP (finetuned): *beige colored building with tan accents and palm trees on both sides of walkway*
BLIP2: *house in villa marina, a villa with many rooms and palm trees*
BLIP2 (finetuned): *a park with lots of trees and benches in front of a large building*
OpenCLIP-CoCa: *residence - villa - maria - di - san - giovanni - near - the - sea - in - taormina*
OpenCLIP-CoCa (finetuned): *a picture of a large building with a bunch of palm trees .*

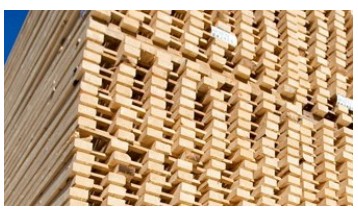

Raw: *3 formas de pedir la mano de tu novia - wikiHow*
BLIP (finetuned): *crates stacked up in a pile on top of each other*
BLIP2: *the building contains five floors of wooden planks*
BLIP2 (finetuned): *a big pile of wooden planks stacked together*
OpenCLIP-CoCa: *the cost of wood pallets*
OpenCLIP-CoCa (finetuned): *a large pile of wooden pallets mounted to a wall .*

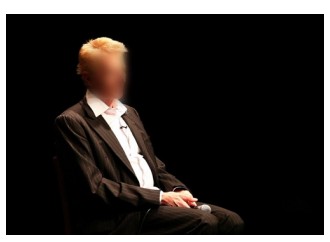

Raw: *lutz*
BLIP (finetuned): *blond haired man in black suit looking at camera*
BLIP2: *a man sitting on a chair with a blank background*
BLIP2 (finetuned): *a man sitting in a chair with a lapel button in front*
OpenCLIP-CoCa: *actor tilda swinton is pictured during a press conference for the film ' a dangerous method ' at the 2 0 1 1 toronto film festival*
OpenCLIP-CoCa (finetuned): *a person sitting on a chair wearing a suit and tie .*

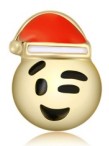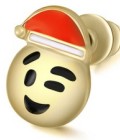

Raw: *Women Personality Creative Christmas Hat Face Expression Gold Earring Funny Cartoon Ear Stud Jewelry Accessories Gift Hot*
BLIP (finetuned): *red and gold tone emojt earring*
BLIP2: *kawaii santa emoticuos en la cabeza*
BLIP2 (finetuned): *a pair of emoji earrings with faces and hats*
OpenCLIP-CoCa: *best christmas gift for her new arrivals emoji earrings christmas emoji earrings*
OpenCLIP-CoCa (finetuned): *a pair of gold earrings with a smiley face and a christmas hat .*

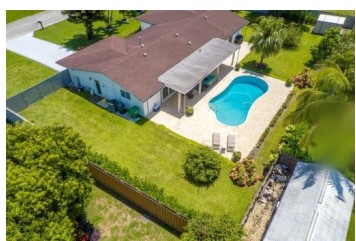

Raw: *10840 SW 126th St photo067*
BLIP (finetuned): *overview of a large backyard with a swimming pool and patio*
BLIP2: *3344 sw 7th st, palm beach*
BLIP2 (finetuned): *a house with a pool from above, with a yard*
OpenCLIP-CoCa: *home for sale in country club shores west palm beach florida*
OpenCLIP-CoCa (finetuned): *aerial image of a pool that has a little bit of shade by the side .*

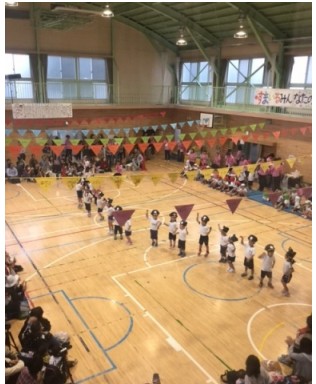

Raw: *image8.JPG*
BLIP (finetuned): *members of a school play soccer in a gymnasium with a crowd*
BLIP2: *a large crowd of kids perform during a dance show*
BLIP2 (finetuned): *a group of young children standing on the basketball court*
OpenCLIP-CoCa: *kid dressed in white standing in a gym area .*
OpenCLIP-CoCa (finetuned): *a group of kids on the gym floor with fans on the floor .*

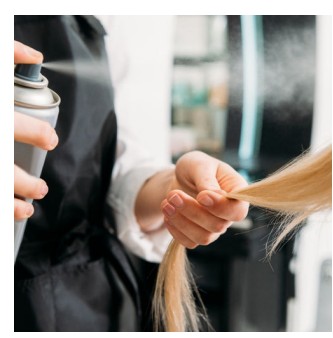

Raw: *hair oil*
BLIP (finetuned): *smiling blonde woman blow drying hair in a salon while getting a mani*
BLIP2: *hair stylist using hair spray in beauty salon*
BLIP2 (finetuned): *a person is using a hairdryer to blow dry a long blonde hair*
OpenCLIP-CoCa: *female hairdresser styling a long blond hair with hairspray in a beauty salon . concept : hair care , hair straightening , hair color correction .*
OpenCLIP-CoCa (finetuned): *a person is spraying a hair dryer on a long blonde hair .*

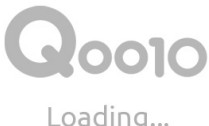

Raw: *Women long sleeve t shirt 2015 Fashion shirts woman Full Comfortable leisure fashion womens long sleeve tops*
BLIP (finetuned): *the qaoo loading logo is shown above the qaoo loading logo*
BLIP2: *qoo10 loading logo on white*
BLIP2 (finetuned): *a picture of an image of a phone screen showing a loading sign*
OpenCLIP-CoCa: *loading _ 1 1 1 1 2 0 _ 0 1 . png*
OpenCLIP-CoCa (finetuned): *a light grey font and a dark grey font with a large white background*

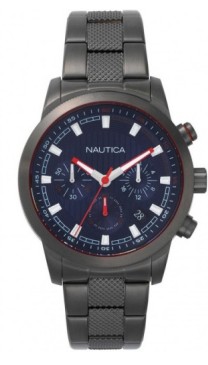

Raw: *Nautica NAPTYR005*
BLIP (finetuned): *navitta mens stainless steel bracelet watch with blue dial*
BLIP2: *nautica men's chronograph watch*
BLIP2 (finetuned): *nautica men's men's chronograph black dial stainless steel bracelet watch*
OpenCLIP-CoCa: *nautica newport chronograph n 2 2 0 0 3 g*
OpenCLIP-CoCa (finetuned): *a mans black watch is shown with red and blue accents*

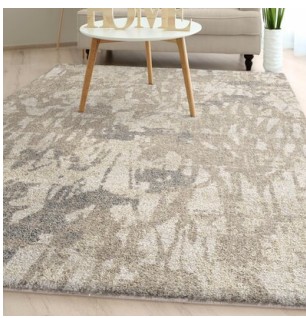

Raw: *Greenberg Weathered Marble Plush Ivory Area Rug*
BLIP (finetuned): *grey rug with a text home on it by a table*
BLIP2: *a grey area rug on a wooden floor*
BLIP2 (finetuned): *a white coffee table with a sign saying home on it. it is sitting on a cream colored rug*
OpenCLIP-CoCa: *rugs and carpets in hyderabad : buy online at best price in ...*
OpenCLIP-CoCa (finetuned): *a rug is shown in a living room with a chair .*

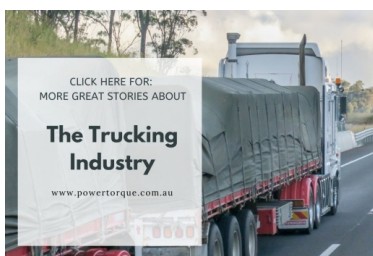

Raw: *productivity, productivity, productivity*
BLIP (finetuned): *drivers guide to the truck industry*
BLIP2: *buy and sell truck parts*
BLIP2 (finetuned): *a white truck with a cover on it drives along a highway*
OpenCLIP-CoCa: *how the trucking industry is changing*
OpenCLIP-CoCa (finetuned): *there are some trucks on the road .*

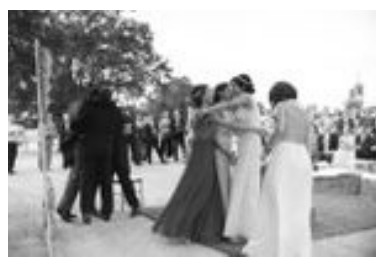

Raw: *Amigas*
BLIP (finetuned): *crowd of people outside a wedding ceremony near several trees*
BLIP2: *a wedding ceremony in the middle of the street*
BLIP2 (finetuned): *a black and white photograph of a number of women in prom dresses*
OpenCLIP-CoCa: *2 0 1 3 0 8 0 5 _ wedding _ carlenan _ 0 0 3*
OpenCLIP-CoCa (finetuned): *a group of people hugging and talking in a group*

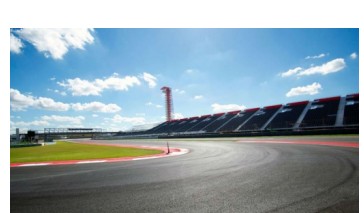

Raw: *Autozone*
BLIP (finetuned): *racing track with a line of seats and a sky background*
BLIP2: *a photo of a grand prix race track, under a blue sky*
BLIP2 (finetuned): *the circuit track is empty, but the sun beams into the sky*
OpenCLIP-CoCa: *circuit of the americas*
OpenCLIP-CoCa (finetuned): *a red and white pole next to a racing track*

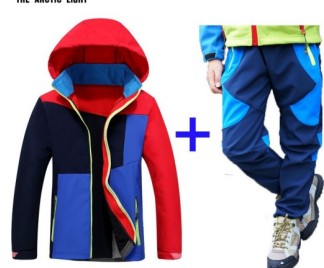

Raw: *Automne hiver enfants manteau et pantalon ensemble capuche veste de Ski et pantalon garçon fille coupe-vent imperméable en plein air camping randonnée*
BLIP (finetuned): *a man wearing a red and blue jacket and a pair of pants and a pair of sneakers*
BLIP2: *the arctic light hooded jacket and pants set*
BLIP2 (finetuned): *the colors of the jacket match the pant color of the jacket*
OpenCLIP-CoCa: *the arctic light 2 0 1 7 children 's clothing sets winter kids ski suits sets windproof waterproof warm jackets coats pants boys set*
OpenCLIP-CoCa (finetuned): *a child standing in their ski wear and a jacket and pants*

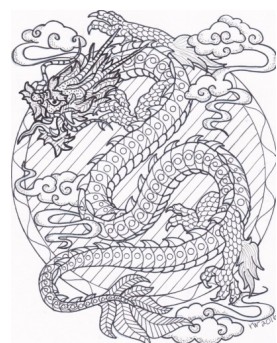

Raw: *1173x1500 Awesome Adult Coloring Pages Printable Zentangle Design*
BLIP (finetuned): *chinese dragon coloring pages dragon coloring pages for adults to print coloring pages*
BLIP2: *dragon coloring pages with large and large dragon*
BLIP2 (finetuned): *a circle with a dragon on it in the center*
OpenCLIP-CoCa: *the 2 5 best chinese dragon drawing ideas on pinterest chinese*
OpenCLIP-CoCa (finetuned): *a chinese dragon looks like a dragon from the movie the karate kid*

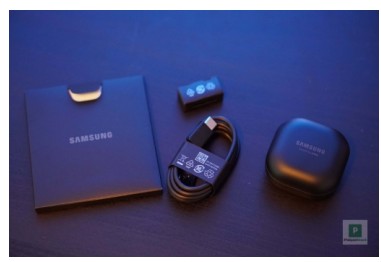

Raw: *Der Lieferumfang*
BLIP (finetuned): *there are several electronics laid out on the table ready to be used*
BLIP2: *samsung galaxy s10e review | a quick tour of the samsung galaxy s10e*
BLIP2 (finetuned): *wireless charging case and remote control, both packaged in the box*
OpenCLIP-CoCa: *best - wireless - chargers - for - samsung - galaxy - note - 8 - s 8 - and - iphone - 8*
OpenCLIP-CoCa (finetuned): *a set of various electronic items sitting on a table .*

# B   Experiment details

Refer to Appendices M and N of the DataComp benchmark [18] for training and evaluation details. To summarize, both `small` and `medium` scales use ViT-B/32 as the image encoder for CLIP, in addition to fixing the hyperparameters used for training: learning rate 5e-4, 500 warmup steps, batch size 4096, AdamW optimizer $\beta_2 = 0.98$. `Large` scale training uses the same hyperparameters, but with batch size 8192 and ViT-B/16 as the image encoder.

Using DataComp infrastructure and the AWS EC2 cloud, a `small` model takes 4 A100 hours to train, while `medium` requires 40 A100 hours and `large` utilizes 960 A100 hours. We additionally report CLIP ViT-L/14 and BLIP2 (OPT 2.7B backbone) inference costs. Recall that we run both of these models on the DataComp's `large` pool to curate the datasets used in this paper. For the CLIP model, we measure throughput at 490 samples per second on a single A100. For BLIP2, we get 75 samples per second on the same hardware. Hence, for the `large` pool of 1.28B samples, we spend 725 A100 hours computing CLIP features and 4,740 A100 hours generating BLIP2 captions.

While the annotation cost (i.e., BLIP2 caption and CLIP score generation) is $6\times$ larger than a single training run proposed by the DataComp benchmark (which is equivalent to going through the entire candidate pool for 1 epoch), this additional cost can be easily amortized with more training epochs over the final training set, as well as with training different downstream models on the improved dataset. For reference, OpenAI trained various CLIP models on the same set of 400M curated image-text pairs; the best performing model was trained on 256 GPUs for 2 weeks, totalling about 86,000 GPU hours [2]. This scale of training is common among existing large vision models. Future work could explore the option of adaptively allocating compute to CLIP training and synthetic caption annotation given a fixed compute budget.

# C   Temperature ablations

| Captioning model | Metric | T=0.5 | T=0.75 | T=1.0 | T=1.5 |
|---|---|---|---|---|---|
| BLIP (finetuned) | ImageNet accuracy | - | 0.207 | **0.212** | - |
| | Average accuracy | - | 0.303 | **0.312** | - |
| BLIP2 | ImageNet accuracy | 0.212 | **0.281** | 0.280 | 0.251 |
| | Average accuracy | 0.300 | **0.357** | 0.353 | 0.332 |
| BLIP2 (finetuned) | ImageNet accuracy | - | 0.227 | **0.234** | 0.221 |
| | Average accuracy | - | 0.325 | **0.326** | 0.311 |
| OpenCLIP-CoCa | ImageNet accuracy | 0.306 | **0.321** | 0.314 | - |
| | Average accuracy | 0.366 | **0.371** | 0.370 | - |
| OpenCLIP-CoCa (finetuned) | ImageNet accuracy | - | 0.252 | **0.264** | 0.262 |
| | Average accuracy | - | 0.364 | **0.374** | 0.364 |

Table 2: Performance on ImageNet and averaged across 38 tasks when training on the captions generated by captioning models in Table 1, with different softmax temperatures. We find that $T = 0.75$ and $T = 1.0$ generally lead to good performance for CLIP training.

# D   More filtering baselines

---

[2]`https://openai.com/research/clip`

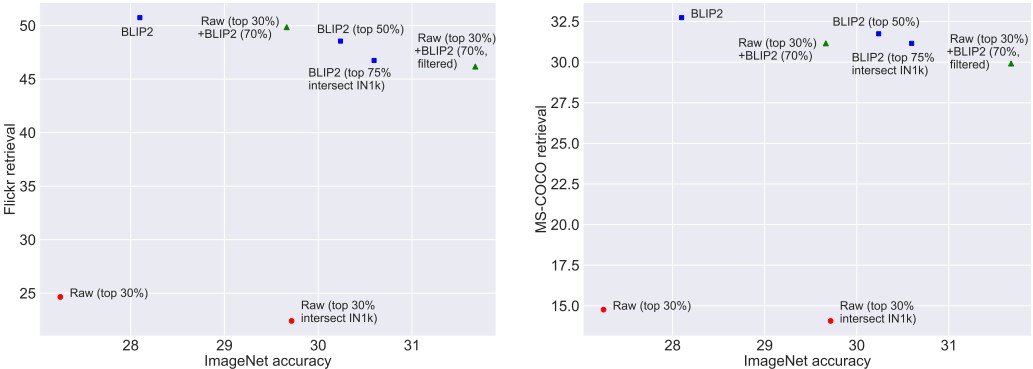

Figure 9: Retrieval performance on Flickr (left) and MS-COCO (right) versus ImageNet accuracy for select baselines. Similar to the findings in Figure 2, we find that using BLIP2 captions or including them in the training data with raw captions significantly boosts performance.

| Baseline | Training set size | ImageNet accuracy | Average accuracy |
|---|---|---|---|
| small scale (12.8M candidate pool, 12.8M training steps) | | | |
| Raw captions (no filtering) | 12.8M* | 0.025* | 0.132* |
| BLIP2 captions (no filtering) | 12.8M | 0.076 | 0.200 |
| Raw captions (top 30%) | 3.8M* | 0.051* | 0.173* |
| BLIP2 captions (top 50%) | 6.4M | **0.080** | **0.203** |
| Raw captions (intersect IN1k and top 30%) | 1.4M* | 0.039* | 0.144* |
| BLIP2 captions (intersect IN1k and top 75%) | 2.4M | 0.073 | 0.192 |
| Raw captions (top 30%) + BLIP2 captions (70%, filtered), intersect IN1k | 2.2M | 0.045 | 0.153 |
| Raw captions (top 30%) + BLIP2 captions (70%, filtered) | 8.4M | 0.076 | 0.197 |
| medium scale (128M candidate pool, 128M training steps) | | | |
| Raw captions (no filtering) | 128M* | 0.176* | 0.258* |
| BLIP2 captions (no filtering) | 128M | 0.281 | 0.357 |
| Top BLIP2 captions across all temperatures (no filtering) | 128M | 0.293 | 0.368 |
| Raw captions (top 30%) | 38M* | 0.273* | 0.328* |
| BLIP2 captions (top 50%) | 64.1M | 0.302 | 0.370 |
| Raw captions (intersect IN1k and top 30%) | 14.0M* | 0.297* | 0.328* |
| BLIP2 captions (intersect IN1k and top 75%) | 23.6M | 0.306 | 0.360 |
| Raw captions (top 30%) + BLIP2 captions (70%, filtered), intersect IN1k | 22.2M | 0.281 | 0.314 |
| Raw captions (top 30%) + BLIP2 captions (70%, filtered) | 83.6M | **0.317** | 0.368 |
| BLIP2 captions (top 50%) + Raw captions (50%, filtered) | 75.3M | 0.310 | **0.376** |
| large scale (1.28B candidate pool, 1.28B training steps) | | | |
| Raw captions (no filtering) | 1.28B* | 0.459* | 0.437* |
| BLIP2 captions (no filtering) | 1.28B | 0.487 | 0.505 |
| Raw captions (top 30%) | 384M* | 0.578* | 0.529* |
| BLIP2 captions (top 50%) | 641M | 0.526 | 0.522 |
| Raw captions (intersect IN1k and top 30%) | 140M* | 0.631* | 0.537* |
| BLIP2 captions (intersect IN1k and top 75%) | 237M | 0.533 | 0.527 |
| Raw captions (top 30%) + BLIP2 captions (70%, filtered), intersect IN1k | 222M | **0.643** | 0.549 |
| Raw captions (top 30%) + BLIP2 captions (70%, filtered) | 834M | 0.598 | **0.551** |

Table 3: Performance for select baselines at small, medium and large scales of DataComp. * indicates numbers obtained from the original paper [18]. Underlined numbers are best-performing baselines from the DataComp benchmark, trained on only raw web-crawled captions. Bolded numbers are the updated state-of-the-art figures after comparing with baselines involving synthetic captions. In general, given a fixed training budget, it is helpful to include more samples in the training pool by carefully replacing noisy raw captions with synthetic captions (i.e., RAW (TOP 30%) + BLIP2 (70%, FILTERED) versus RAW (TOP 30%)). We experiment with many more filtering and mixing methods at the medium scale and report how the performance varies with CLIP score filtering threshold, see Table 4.

| CLIP score filtering | 10% | 20% | 30% | 50% | 75% | 90% |
|---|---|---|---|---|---|---|
| Cosine similarity threshold | | | | | | |
| Raw captions | 0.295 | 0.266 | 0.243 | 0.203 | 0.160 | 0.129 |
| BLIP2 captions | 0.315 | 0.292 | 0.277 | 0.251 | 0.217 | 0.187 |
| Only raw captions | | | | | | |
| Training set size | 12.8M* | 25.7M* | 38.4M* | 64.1M* | 96.1M* | 115M* |
| ImageNet accuracy | 0.198* | 0.260* | 0.273* | 0.254* | 0.212* | 0.188* |
| Average accuracy | 0.277* | 0.322* | 0.328* | 0.315* | 0.285* | 0.266* |
| Only BLIP2 captions | | | | | | |
| Training set size | 12.8M | 25.6M | 38.5M | 64.1M | 96.0M | 115M |
| ImageNet accuracy | 0.146 | 0.249 | 0.275 | 0.302 | 0.300 | 0.293 |
| Average accuracy | 0.254 | 0.333 | 0.356 | 0.370 | 0.365 | 0.366 |
| Only BLIP2 captions, for top % based on cosine similarity of image and *raw* text | | | | | | |
| Training set size | 12.8M | 25.7M | 38.4M | 64.1M | 96.1M | 115M |
| ImageNet accuracy | 0.192 | 0.245 | 0.261 | 0.266 | 0.267 | 0.276 |
| Average accuracy | 0.280 | 0.330 | 0.346 | 0.342 | 0.349 | 0.356 |
| Raw captions for top % + BLIP2 captions for the remaining examples | | | | | | |
| Training set size | 128M | 128M | 128M | 128M | 128M | 128M |
| ImageNet accuracy | 0.286 | 0.296 | 0.297 | 0.286 | 0.250 | 0.215 |
| Average accuracy | 0.360 | 0.357 | 0.365 | 0.349 | 0.323 | 0.293 |
| Raw captions for top % + BLIP2 captions for the remaining examples, subject to the same cosine similarity threshold | | | | | | |
| Training set size | 30.5M | 59.5M | 83.6M | 114M | 127M | 128M |
| ImageNet accuracy | 0.267 | 0.310 | **0.317** | 0.296 | 0.251 | 0.212 |
| Average accuracy | 0.343 | 0.372 | 0.368 | 0.352 | 0.313 | 0.285 |
| BLIP2 captions for top % + raw captions for the remaining examples, subject to the same cosine similarity threshold | | | | | | |
| Training set size | 17.1M | 32.8M | 47.7M | 75.3M | 105M | 121M |
| ImageNet accuracy | 0.212 | 0.272 | 0.298 | 0.310 | 0.298 | 0.285 |
| Average accuracy | 0.305 | 0.353 | 0.367 | **0.376** | 0.375 | 0.355 |
| Concatenate raw & BLIP2 captions for top % + BLIP2 captions for the remaining examples, subject to the same cosine similarity threshold | | | | | | |
| Training set size | 30.5M | 59.5M | 83.6M | 114M | 127M | 128M |
| ImageNet accuracy | 0.250 | 0.287 | 0.299 | 0.286 | 0.269 | 0.262 |
| Average accuracy | 0.336 | 0.368 | 0.367 | 0.359 | 0.340 | 0.337 |
| Top % raw captions + top % BLIP2 captions | | | | | | |
| Training set size | 25.6M | 51.3M | 76.9M | 128M | - | - |
| ImageNet accuracy | 0.238 | 0.285 | 0.297 | 0.300 | - | - |
| Average accuracy | 0.318 | 0.358 | 0.366 | 0.356 | - | - |
| BLIP2 captions - top % intersect with examples from IN1k clustering | | | | | | |
| Training set size | - | - | 10.0M | 16.4M | 23.6M | 27.1M |
| ImageNet accuracy | - | - | 0.243 | 0.289 | 0.306 | 0.301 |
| Average accuracy | - | - | 0.310 | 0.343 | 0.360 | 0.344 |

Table 4: Summary of how various filtering and mixing strategies perform on ImageNet and on average across 38 evaluation tasks in DataComp, given a 128M candidate pool (`medium` scale). * indicates numbers obtained from Gadre et al. [18]. Note that all resulting training sets are trained for a fixed number of steps (128M samples seen) and other training variables (e.g., architecture, hyperparameters) are kept constant. Synthetic captions are generated using pre-trained BLIP2 model with top-K sampling (K = 50) and softmax temperature 0.75. We find that at this scale, approaches that yield the best ImageNet and average accuracies leverage a combination of raw and synthetic captions.

# E  Synthetic caption analysis

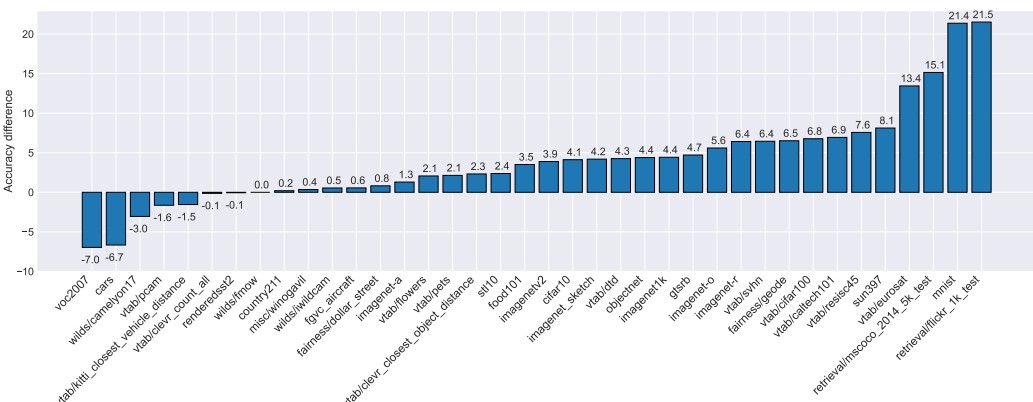

Figure 10: **We find that expanding a training set of filtered raw data by using BLIP2 captions for some of the discarded images improves performance on 30 out of 38 evaluation tasks, in addition to boosting average accuracy by 4%.** We compare performance on each task between training on top 30% of examples with raw captions (based on CLIP score) and training on the same set of examples but with the addition of BLIP2 captions for the remaining 70% images, filtered by the same CLIP score threshold. In Table 3, we have shown that adding BLIP2 captions improves ImageNet accuracy by 4.4% and average accuracy by 4%. With this breakdown, we find that the performance improvement applies to most of the tasks in the evaluation set, especially retrieval.

We investigate whether there are systematic differences in training with raw and generated text when it comes to recognizing certain object categories. To do so, we examine two CLIP models that perform similarly on ImageNet (i.e., $\pm 0.2\%$): one trained on only raw captions and one trained on only generated captions, both training sets have been filtered with CLIP score ranking to select the top 30% image-text pairs. In Figure 11, we analyze performance on each ImageNet class, categorized as either 'living' or 'non-living' thing based on where the classname synset is located in the WordNet hierarchy. We observe that class-wise classification performance is scattered evenly around the $y = x$ line, indicating that compared to web-crawled captions, synthetic captions do not exhibit a particular disadvantage on either 'living' or 'non-living' concepts.

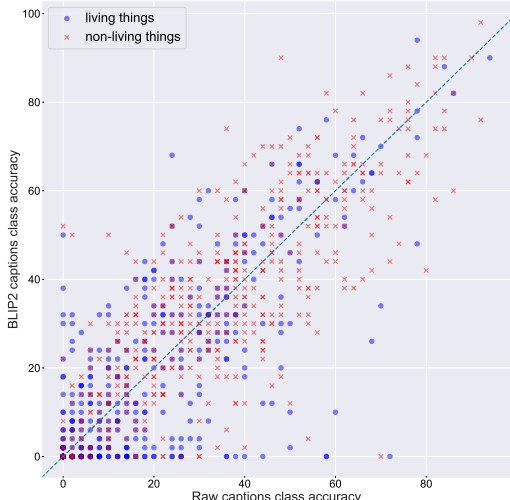

Figure 11: We break down per-class performance on ImageNet, between a CLIP model trained on only raw captions and one trained on only synthetic captions with similar overall ImageNet accuracy. We find no systematic trends in the performance of either model when it comes to classifying 'living' or 'non-living' things.

# F    Performance at Scale

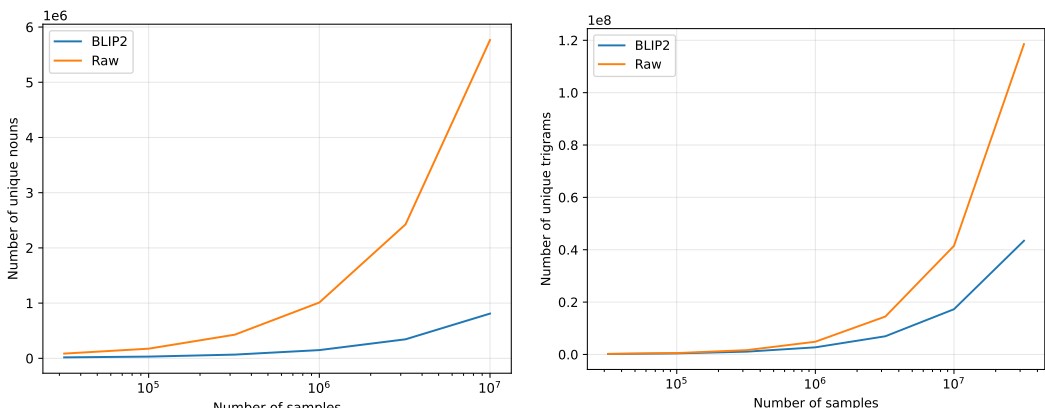

Figure 12: **Our simple analyses of text properties suggest that the text diversity provided by synthetic captions may not scale as well as that of raw captions scraped from the Internet.** We measure the number of unique nouns and unique trigrams for random subsets of BLIP2 and raw captions of various sizes. We observe that on both metrics, the scaling trend for synthetic captions is worse than that of raw captions. This increasing gap in data diversity may impact the performance benefits we can expect to obtain from using synthetic captions, when dealing with a larger scale of training data.

# G    Experiments with LAION-COCO

Our experiments with synthetic captions are partly inspired by the release of LAION-COCO dataset [45], which used BLIP [29] with various hyperparameter settings to caption LAION-5B data [46], and then selected the top synthetic caption for each image based on the cosine similarity output by OpenAI's CLIPs [40]. We pick a random set of 100M samples from LAION-COCO and train on this set using DataComp's `medium` scale configuration (i.e., 128M steps), with either only the raw captions or only the top BLIP captions that come with the dataset. We find that training on BLIP captions significantly lags behind training on raw captions, measured by both ImageNet and average accuracies (Figure 13). Consequently, a natural question is how much of this gap can be overcome with progress in image captioning models, e.g. the release of BLIP2.

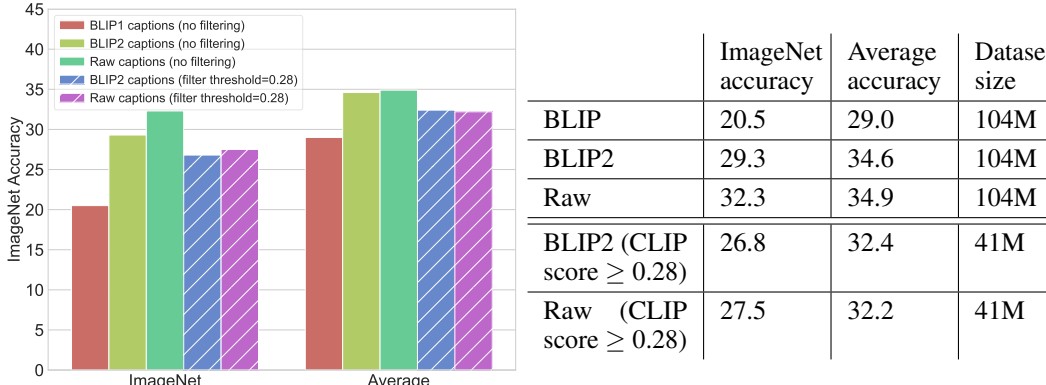

| | ImageNet accuracy | Average accuracy | Dataset size |
|---|---|---|---|
| BLIP | 20.5 | 29.0 | 104M |
| BLIP2 | 29.3 | 34.6 | 104M |
| Raw | 32.3 | 34.9 | 104M |
| BLIP2 (CLIP score $\geq$ 0.28) | 26.8 | 32.4 | 41M |
| Raw (CLIP score $\geq$ 0.28) | 27.5 | 32.2 | 41M |

Figure 13: **BLIP2 significantly closes the performance gap between BLIP captions and raw captions on LAION-COCO; when controlled for noise level, the performance difference between using BLIP2 and using raw captions is actually negligible.** We use BLIP2 [30] to generate captions for 100M random samples from the LAION-COCO dataset [45], which already come with corresponding BLIP [29] captions. We find that advances in the BLIP model family help synthetic captions close the gap with raw captions, measured by the zero-shot performance of CLIP trained on the captions. After applying a cosine similarity threshold of 0.28 to the BLIP2 training pool, just like how LAION data was originally selected, we find that using either raw captions or synthetic captions for the resulting set of examples makes little difference (hatched columns).

We proceed to generating BLIP2 captions for the same set of 100M images, using only one configuration from the original hyperparameter grid in [45] due to compute constraints. Despite the lack of tuning, the new BLIP2 captions manage to close the previous ImageNet performance gap by 75% and come close to the average accuracy obtained from training on raw captions (see table in Figure 13). Since raw data in LAION was already filtered with a CLIP score threshold of 0.28 during the dataset construction, we next experiment with applying the same filtering to BLIP2 captions, in order to control for noise quality in the caption data. On the resulting 41M images, using BLIP2 captions is about as effective as using raw captions (-0.7% ImageNet accuracy and +0.2% average accuracy).

We note that LAION is considered a curated web dataset, with heavy cosine similarity filtering being one of the preprocessing steps. This in turn leads to approximately 90% of the raw data from Common Crawl to be discarded, according to Schuhmann et al. [46]. Since LAION only retains about 10% of the original candidate pool, similar experiments in DataComp [18] have shown that further CLIP score filtering on these top examples will only hurt performance. In addition, given that the selected raw captions are already relatively clean (measured via image-text cosine similarity), and there is no record of datapoints that were filtered out for further experimentation, we find LAION-COCO to be an unsuitable benchmark for studying the utility of synthetic captions. Our experiments here mainly seek to demonstrate that progress in image captioning models (e.g., the BLIP model family) can translate to better text supervision for CLIP training that rivals the effectiveness of using raw captions.

## H  Fairness implications of using synthetic captions

We examine zero-shot classification accuracy of predicting race and gender from face images in the Fairface dataset [26], for a model trained on only filtered raw captions, one trained on only filtered synthetic captions, and one trained on both. We acknowledge that there are limitations to these evaluations as race and gender should not be considered fixed categories.

With Fairface, we find that using synthetic captions improves the classification performance on the disadvantaged group (e.g. female) significantly, and reduces the performance gap between male and female groups while still boosting the overall performance on all race categories. We leave more extensive study of the fairness implications of using synthetic data (including and beyond gender biases) to future work.

| Gender | Model | Race | | | | | | |
|---|---|---|---|---|---|---|---|---|
| | | Black | White | Indian | Latino/ Hispanic | Middle Eastern | South East Asian | East Asian |
| Male | Raw (top 30%) | 93.0 | 88.8 | 91.2 | 90.8 | 92.3 | 85.3 | 81.3 |
| | BLIP2 (top 30%) | 87.2 | 73.7 | 77.2 | 74.9 | 78.6 | 72.0 | 64.0 |
| | Raw (top 30%) + BLIP2 (70%, filtered) | 90.5 | 75.0 | 79.7 | 79.4 | 81.1 | 72.4 | 65.3 |
| Female | Raw (top 30%) | 20.3 | 47.1 | 35.1 | 42.0 | 40.9 | 44.9 | 56.8 |
| | BLIP2 (top 30%) | 36.9 | 70.8 | 57.9 | 67.5 | 67.4 | 64.1 | 78.4 |
| | Raw (top 30%) + BLIP2 (70%, filtered) | 32.9 | 74.8 | 56.5 | 66.3 | 67.9 | 67.8 | 81.9 |
| Overall | Raw (top 30%) | 56.7 | 68.0 | 63.2 | 66.4 | 66.6 | 65.1 | 69.1 |
| | BLIP2 (top 30%) | 62.1 | 72.3 | 67.6 | 71.2 | 73.0 | 68.1 | 71.2 |
| | Raw (top 30%) + BLIP2 (70%, filtered) | 61.7 | 74.9 | 68.1 | 72.9 | 74.5 | 70.1 | 73.6 |

Table 5: Using synthetic captions in the training mix improves classification performance on Fairface for the minority group (i.e. female) across all race categories.

