# OpenReview forum: "Improving multimodal datasets with image captioning"
_NeurIPS.cc/2023/Track/Datasets_and_Benchmarks — NeurIPS 2023 Datasets and Benchmarks Poster_

### Official Review · Reviewer_DC33 · 2023-07-19
**Review: Improving multimodal datasets with image captioning**

**Rating:** 6
**Confidence:** 4
**Correctness:** The claims seem sufficient and correct.

**Strengths:**

This is an interesting paper which provides an interesting insight into how training CLIP-like models for pre-training can be improved with data filtering. The results are convincing and strong, and show significant benefits over baseline approaches. The paper has a lot of good analysis evaluating some of the hypotheses, and the included details are helpful and insightful. The approach, while lacking some novelty (see related work), is a valuable contribution to the field, as it clearly demonstrates the effect of pre-training on synthetic captions vs. alt-text (or a mix of synthetic and alt-text). The problem of building more efficient datasets from existing collected data is also an interesting area, which may be applicable outside of the field.

**Additional Feedback:**

Overall, I'm relatively split on this paper - my primary concern being the fit with the track. While the paper results are interesting, and demonstrate some interesting results on synthetic vs. real data, I wonder if they are general enough to be of interests in the datasets/benchmarks track, particularly without releasing the filtered data. If the filtered data were released, I would have fewer track concerns, as the community would be able to benefit not just from the process of data filtering, but also the updated fine-tuning datasets. The paper is overall relatively solid, and fairly interesting.

**Clarity:**

The paper is clear, and detailed. I'm confident that I could reproduce the results in this paper given the information in the paper.

**Documentation:**

No documentation (no new dataset).

**Ethics:**

No ethics concerns.

**Limitations:**

The discussion of limitations exists, but is somewhat limited. It references a key detail: toxic information and harmful stereotypes in alt-text data, however this is not a core limitation of the approach itself, In fact, this approach may help to limit toxic information produced by pre-training models, as the captioning models are often less likely to produce toxic information (usually because of explicit training or supervision) than the models themselves.

It would be nice to see a more detailed discussion of the limitations of the method itself. Some potential discussion ideas are:
- Is there a risk of reducing the dataset coverage by filtering?
- Does CLIP filtering the data, or using synthetic captions induce (or reinforce) bias in the underlying dataset? (for example, it's well known that captioning models are often gender-impaired, does synthetic caption generation reinforce a gender biases, and cause imbalances in the downstream CLIP predictions?)
- Does training on synthetic captions limit generalization ability of the pre-trained models, particularly to tasks that are not classification/captioning centric?
- What does the computational overhead look like? Are there costs to using synthetic data?


**Opportunities For Improvement:**

While the paper is interesting, and has some strong results, there are also several areas that could be improved.

**Track Fit**: It's not entirely clear to me that this paper fits in the datasets and benchmarks track. The paper is primarily an investigation of using synthetic large-scale data compared to the real data itself, as an investigation of how underlying data caption quality impacts training performance of downstream models. The experiments are only performed on a single dataset, with a single downstream training architecture, and while the results are interesting, I'm not convinced that it fits with the "datasets and benchmarks" theme. It would help if the authors aimed to release the captioning data (which I couldn't tell would happen), in which case, it fits much closer, but I am somewhat concerned about overall fit.

**Over-specialization**: While the pre-training results lead to strong performance in the downstream tasks of image classification, I'm curious if the choice of captioning model architectures (namely ViT + BLIP) is leaking information about the downstream task during pre-training. Because both ViT and BLIP are pre-trained on ImageNet-21K, and specialized for classification, it seems natural to me that BLIP captions would encode more information necessary for classification than a standard alt-text. I would worry that this leads to degradation in performance across other downstream tasks (perhaps open-domain image captioning, or VQA), as the models are implicitly focusing only on relevant image details. While this is still acceptable (given that the task is image classification), I think that this "self-distillation" effect is worth discussing more in the text. Similarly, I wonder if the language distribution itself contains some amount of information about recall/classification tasks which is helpful - it would be interesting to discuss this further.

**Clip Similarity vs. Image-Text Alignment**: The notion that is present throughout the work is that clip similarity is equivalent to image-text alignment is somewhat misleading. While in many cases  high clip similarity implies high image-text aligment, it is not always the case (the CLIP model can perform poorly, image-text alignment may be non-visual, etc.). It may be best to soften some of the language around this, particularly in the claims in Figure 4 -- just because the distribution of image-text cosine similarity scores has a lower mean, does not really mean that the alt-text is poorly aligned with the image, it does, however, indicate that there are axes of image-alt-text alignment that are not captured by the CLIP model. Similarly with Figure 3 - the fact that tokens are more visual does not necessarily mean that there is more "information" - rather, it means that more visual tokens are generated. This should be made explicit in the paper to avoid confusion.

**Impact of Filtering**: It would be nice to see some aditional analysis of Figure 2 (b), which discusses how filtering percentage impacts the performance of the models. It seems like filtering is always important and is a main contribution of the paper, however it seems like Figure 2 is under-represented in the text, and discussed somewhat superficially.

**Conflicting notions of diversity:** On L45, a motivation of the paper is that while existing data filtering methods are effective at reducing noise, they often hurt the diversity of the original training data in the process. Synthetic captions can help address this limitation, however a lot of the paper discusses how synthetic captions are actually less diverse than raw alt-text. It might be useful to clarify the motivation on L45.

**Surprising results:** I don't know if the claim on L48 is that remarkable: while better performance on image captioning benchmarks is good for captioning models, the task of training a CLIP model is much different, and while the tasks are certainly related, I'm not sure that it's that surprising that they don't transfer 1-1. Similarly, the claims that increased diversity are correlated with increased model performance are not exactly revolutionary, and are heavily implied by work on dataset core-sets, active learning, and other dataset filtering approaches. It might be best to reword some of the paragraphs to indicate this.

**Definitions:** On L264 (and throughout the paper), the paper refers to "noise level" and "text diversity" as being separate axes - however the concept of "noise" is never explicitly defined. It would be good to have an explicit definition somewhere in the paper. On L270, the term "Image quality" is used, but that term is never explicitly defined.

**Relation To Prior Work:**

While the related work is good, and the paper is novel, it is worth noting that similar ideas without the filtering have been explored in [1 (13 in the paper)], where they show that dense image annotations (such as caption) are better for pre-training than sparse annotations. While [1] was cited, it would be nice to see a good discussion of the key differences. A similar idea (using captioning models for pre-training) was also explored in video description in both [2,3], without the filtering, where image captions are aligned with video for data augmentation.

A related field is Symbolic Knowledge Distillation [4,5,6], which was not mentioned, but discusses iterated filtering/training in the text domain.

It might also be interesting to discuss captioning models that focus on more descriptive captions, such as [7,8], which may have much better pre-training behavior - but this is not necessary or required.

The claim on L163 (and generally discussed through the paper):  "Notably, captioning models that are only pre-trained have very poor CIDEr scores; going with this metric would have suggested that these models are not suitable for caption generation at all." has been well investigated by [8, 9, 10] among others.

[1/13] Desai, Karan, and Justin Johnson. "Virtex: Learning visual representations from textual annotations." Proceedings of the IEEE/CVF conference on computer vision and pattern recognition. 2021.

[2] Arsha Nagrani, Paul Hongsuck Seo, Bryan Seybold, Anja Hauth, Santiago Manén, Chen Sun, and Cordelia Schmid. Learning audio-video modalities from image captions. ArXiv, abs/2204.00679, 2022.

[3] Lialin, Vladislav, et al. "Scalable and Accurate Self-supervised Multimodal Representation Learning without Aligned Video and Text Data." Proceedings of the IEEE/CVF Winter Conference on Applications of Computer Vision. 2023.

[4] Bhakthavatsalam, Sumithra, Chloe Anastasiades, and Peter Clark. "Genericskb: A knowledge base of generic statements." arXiv preprint arXiv:2005.00660 (2020).

[5] West, Peter, et al. "Symbolic Knowledge Distillation: from General Language Models to Commonsense Models." Proceedings of the 2022 Conference of the North American Chapter of the Association for Computational Linguistics: Human Language Technologies. 2022.

[6] Bhagavatula, Chandra, et al. "I2d2: Inductive knowledge distillation with neurologic and self-imitation." arXiv preprint arXiv:2212.09246 (2022).

[7] Zhu, Deyao, et al. "Chatgpt asks, blip-2 answers: Automatic questioning towards enriched visual descriptions." arXiv preprint arXiv:2303.06594 (2023).

[8] Chan, David M., et al. "IC3: Image Captioning by Committee Consensus." arXiv preprint arXiv:2302.01328 (2023).

[9] Caglayan, Ozan, Pranava Madhyastha, and Lucia Specia. "Curious case of language generation evaluation metrics: A cautionary tale." arXiv preprint arXiv:2010.13588 (2020).

[10] Chan, David M., et al. "What's in a Caption? Dataset-Specific Linguistic Diversity and Its Effect on Visual Description Models and Metrics." Proceedings of the IEEE/CVF Conference on Computer Vision and Pattern Recognition. 2022.

**Summary And Contributions:**

This paper investigates the use of synthetic captions for pre-training CLIP-based models instead of alt-text on large-scale datasets. Instead of training on raw alt-text, the paper explored training CLIP-based models on synthetic captions, filtered for caption-image similarity, generated by image-captioning models. The results are clear: using synthetic captions improves the performance of retrieval methods on MS-COCO and Flickr Datasets. Further, analysis of the synthetic captions demonstrates some surprising effects including that substantial gains on retrieval tasks can be obtained solely by using better aligned captions and image quality becomes increasingly important at larger scales.

---

> ### Author Response · Authors · 2023-08-17
> **Response to Reviewer DC33**
>
> We thank the reviewer for taking the time to offer a thorough review of our work!
>
> **Track Fit.** Under the call for papers for this track, we believe our work is in line with the “Data-centric AI methods and tools, e.g. to measure and improve data quality or utility, or studies in data-centric AI that bring important new insight” direction. We offer some examples of accepted Datasets & Benchmarks papers from past years that also seek to improve existing datasets: ​​[ImageNet-21K Pretraining for the Masses (Ridnik et al., 2021)](https://datasets-benchmarks-proceedings.neurips.cc/paper/2021/hash/98f13708210194c475687be6106a3b84-Abstract-round1.html), [SynthBio: A Case Study in Faster Curation of Text Datasets (Yuan et al., 2021)](https://datasets-benchmarks-proceedings.neurips.cc/paper/2021/hash/9fc3d7152ba9336a670e36d0ed79bc43-Abstract-round2.html), and [Finding Naturally Occurring Physical Backdoors in Image Datasets (Wenger et al., 2022)](https://proceedings.neurips.cc/paper_files/paper/2022/hash/8af749935131cc8ea5dae4f6d8cdb304-Abstract-Datasets_and_Benchmarks.html).
>
> **Data release. NOTE: the links we provide below are not anonymized.** We have provided [a variety of BLIP2 captions](https://huggingface.co/datasets/thaottn/DataComp_medium_pool_BLIP2_captions) generated with 5 different softmax temperatures for the 128M images in the medium pool. We are in the process of releasing [1.28B BLIP2 captions](https://huggingface.co/datasets/thaottn/DataComp_large_pool_BLIP2_captions) corresponding to the images in the large pool of DataComp.
>
>
> **“It seems natural to me that BLIP captions would encode more information necessary for classification than a standard alt-text.”** We respectfully disagree with this point. We have experimented with using the original BLIP model (note: different from BLIP2) to augment CommonPool as well as LAION data (see Section G for LAION-COCO experiments), and observed that the generated captions are not specific enough to yield competitive zero-shot performance on ImageNet compared to just filtering raw data. We again emphasize that choosing a captioning model to effectively augment existing image-text datasets and attain state-of-the-art performance is non-trivial. In addition, we also find that at medium and large scales, the best filtering and mixing strategies involve *both* raw and synthetic captions, rather than relying on only either.
>
>
> **“I would worry that this leads to degradation in performance across other downstream tasks (perhaps open-domain image captioning, or VQA), as the models are implicitly focusing only on relevant image details”/ “Does training on synthetic captions limit generalization ability of the pre-trained models, particularly to tasks that are not classification/captioning centric?”** Our work focuses on evaluating zero-shot generalization behaviors of CLIP, so open-domain image captioning would not be suitable. Given your concern, however, we have performed **new experiments with an additional task**, visual question answering, using the VQA v1 dataset (Antol et al., 2015). We find that using BLIP2 captions in the training data consistently yields improvement on this task: (i) our best baseline using both sources of captions, Raw (top 30%) + BLIP2 (70%, filtered), gets 0.0712, while the best baseline using raw captions from the medium scale, Raw (top 30% intersect IN1k), scores 0.0573, (ii) despite yielding similar ImageNet performance, a model trained on only top 30% synthetic captions outperforms a model trained on only top 30% raw captions on this task (0.0787 vs. 0.0652). We note that the low performance on VQA tasks is expected for CLIP trained on only image-text pairs, as demonstrated in previous work [1, 2].
>
> [1] How Much Can CLIP Benefit Vision-and-Language Tasks? Shen et al., 2021.
>
> [2] OBELISC: An Open Web-Scale Filtered Dataset of Interleaved Image-Text Documents. Laurencon et al., 2023.
>
>
> **“While in many cases high clip similarity implies high image-text aligment, it is not always the case.”** This is a fair point, we will clarify in the paper that for the lack of precise metrics, the use of CLIP similarity and grounding ratio serves as heuristics for capturing image-text alignment and information conveyed by the caption.
>
>
> **Impact of Filtering.** Thank you for the pointer. We will add more discussion of Figure 2(b) in the main text. In general, filtering is indeed important, but CLIP similarity filtering has also been commonly used in existing work, so we did not want to over-emphasize that as our main contribution. An important finding that we chose to discuss in detail instead is that even when augmenting existing training data with synthetic captions, it is still necessary to filter the new image-generated-text pairs to the same degree as what was done on raw data.

---

> > ### Author Response · Authors · 2023-08-17
> > **Response to Reviewer DC33 (cont.)**
> >
> > **Conflicting notions of diversity.** We agree that further clarification on L45 is needed to avoid confusion. We did not intend for the diversity discussions to be conflicting. While it is true that synthetic captions are overall less diverse than raw alt-text, we find that synthetic captions are still effective at complementing raw captions and improving the number of useful training samples, especially when raw data is often heavily filtered in existing work. Our results seek to caution against relying on only synthetic captions, especially at larger training regimes, as there is still room for improvement in terms of linguistic diversity for synthetic captions to match web-crawled text.
> >
> >
> > **“Surprising results. I don't know if the claim on L48 is that remarkable... Similarly, the claims that increased diversity are correlated with increased model performance are not exactly revolutionary.”** First, it is true that the image captioning and CLIP training tasks are related but do not necessarily transfer 1-1. We posit that given a range of captioning models to choose from to generate more relevant captions, it is likely that practitioners will opt for the more specialized version (i.e. having been fine-tuned on some related task) rather than the generic model pre-trained on noisy web data. In fact, LAION-COCO uses BLIP fine-tuned for image captioning to generate their synthetic captions (see their [code](https://github.com/andreaskoepf/laion_idle_cap/tree/main/docker)). In Section 4, we wanted to point out that model specialization is not necessarily good for the purpose of generating text supervision for CLIP.
> >
> > Secondly, we did not claim that increased diversity being correlated with increased model performance is a contribution of our work. We only reference increased diversity as a motivation for adding synthetic captions to existing filtered datasets, as well as to “offer some intuition for our best baseline uncovered in Section 5”.
> >
> >
> > **Definitions.** We will add more clarification for what we mean by “noise” and “image quality”. For the latter, we use embedding distance to the ImageNet-1K training set as a heuristic for selecting more relevant images.
> >
> >
> > **“Is there a risk of reducing the dataset coverage by filtering?”** This is one of the main motivations of our use of synthetic captions, as discussed in the Introduction (Lines 44-45): while many existing work finds that data filtering is crucial for getting good performance (as the raw web data can be incredibly noisy), the process probably also removes a lot of useful concepts that are captured in the discarded images. Thus, it is important to balance the goals of maintaining good data coverage while keeping the noise level low (the latter is approximated by image-text similarity threshold), and synthetic captions can help more training samples from the candidate pool fit these criteria.
> >
> > **“Does CLIP filtering the data, or using synthetic captions induce (or reinforce) bias in the underlying dataset? (for example, it's well known that captioning models are often gender-impaired, does synthetic caption generation reinforce gender biases?)”** This is what we meant by saying in the Conclusion that generated captions inherit certain biases from the captioning models, and using these captions for training can amplify the biases. Per your suggestion, we examine zero-shot classification accuracy of predicting race and gender from face images in the Fairface dataset (Karkkainen et al., 2021), for a model trained on only filtered raw captions, one trained on only filtered synthetic captions, and one trained on both. We acknowledge that there are limitations to these evaluations as race and gender should not be considered fixed categories.
> >
> > With Fairface, we find that using synthetic captions improves the performance of the disadvantaged group (e.g. female) significantly, and reduces the performance gap between males and females while still boosting the overall performance on all race categories. In general, we believe that studying the fairness implications of using synthetic data (including and beyond gender biases) is a highly important direction for future work, and we will include more fairness analysis in our paper.
> > |Gender|Model|Race| | | | | | |
> > |:----|:----|:----|:----|:----|:----|:----|:----|:----|
> > | | |Black|White|Indian|Latino/ Hispanic|Middle Eastern|South East Asian|East Asian|
> > |Male|Raw (top 30%)|93.0|88.8|91.2|90.8|92.3|85.3|81.3|
> > | |BLIP2 (top 30%)|87.2|73.7|77.2|74.9|78.6|72.0|64.0|
> > | |Raw (top 30%) + BLIP2 (70%, filtered)|90.5|75|79.7|79.4|81.1|72.4|65.3|
> > |Female|Raw (top 30%)|20.3|47.1|35.1|42.0|40.9|44.9|56.8|
> > | |BLIP2 (top 30%)|36.9|70.8|57.9|67.5|67.4|64.1|78.4|
> > | |Raw (top 30%) + BLIP2 (70%, filtered)|32.9|74.8|56.5|66.3|67.9|67.8|81.9|
> > |Overall|Raw (top 30%)|56.7|68|63.2|66.4|66.6|65.1|69.1|
> > | |BLIP2 (top 30%)|62.1|72.3|67.6|71.2|73.0|68.1|71.2|
> > | |Raw (top 30%) + BLIP2 (70%, filtered)|61.7|74.9|68.1|72.9|74.5|70.1|73.6|

---

> > > ### Author Response · Authors · 2023-08-17
> > > **Response to Reviewer DC33 (cont.)**
> > >
> > > **“What does the computational overhead look like? Are there costs to using synthetic data?”** Please refer to Appendix Section B for a discussion of the computational overhead. To summarize, we find that in the context of the DataComp benchmark, in which the number of training steps is the same as the size of the candidate pool, the computational overhead is relatively significant. However, assuming the resulting filtered dataset is then trained on for multiple epochs and used for multiple models (as in the case of OpenAI's CLIPs), the cost of caption generation and filtering would be sufficiently amortized.
> > >
> > >
> > > **Related Work.** Thank you for the references. Due to space constraints, we could not discuss the differences between our work and certain cited work in detail. We will revise the related work section taking your suggestions into account.

---

> > > > ### Comment · Reviewer_DC33 · 2023-08-21
> > > > **Thanks for the responses**
> > > >
> > > > Thanks for the responses! I appreciate the authors taking the time to include a detailed discussion of the results. While I stand by my original numeric score (6), as I am concerned that the authors may not be able to incorporate all of the updated details/experiments, I believe that if included, these details add significantly to the work, and I believe this paper is a good candidate for acceptance.
> > > >
> > > > I do still have concerns about track fit (which is primarily why my score is a 6 and not a 7), but I leave that to the AC to decide.

---

### Official Review · Reviewer_fUUj · 2023-07-21

**Rating:** 8
**Confidence:** 3
**Correctness:** The claims appear to be correct.
**Clarity:** The paper is very well written.

**Strengths:**

- Paper is clearly written and easy to understand.
- Strong experiment results with a fresh angle. Synthetic captions have been around for a while, but this is one of the few works that attempt to formally study it, and it has done an elegant job.
- Large-scale experiments that support the claim.

**Additional Feedback:**

How would multilingualism affect (or not affect) the analysis?

**Documentation:**

The details appear to be sufficient to reproduce the results.

**Ethics:**

There seems to be no obvious ethical concerns.

**Limitations:**

The authors have adequately addressed the limitations and potential negative impacts of their work.

**Opportunities For Improvement:**

- The author may want to give names to each mixing strategy to make the paper more easily referrable from future work.

**Relation To Prior Work:**

The manuscript has clearly discussed the difference between this work and its predecessors.

**Summary And Contributions:**

The manuscript explores how synthetic captions, generated from image captioning models, could improve the quality of image-text-pair datasets, which, scraped from the internet, contains a large amount of noise. Through several strategies that mix synthetic captions with web-scraped captions on scales up to 1.2B image-text pairs, the authors have demonstrated state-of-the-art performance on the DataComp benchmark. Finally, the authors investigate the source of effectiveness of the image captioning models and draw a relationship between a model's performance on image captioning benchmarks and the value of the synthetic caption it generates.

---

> ### Author Response · Authors · 2023-08-17
> **Response to Reviewer fUUj**
>
> We thank the reviewer for their time and for recognizing the strengths of our paper!
>
> **“Give names to each mixing strategy to make the paper more easily referrable from future work.”** Thank you for this great suggestion. We will take this into consideration and revise the paper accordingly.
>
> **“How would multilingualism affect (or not affect) the analysis?”** It is likely that our current training setup would lead to a performance drop when evaluating on prompts in other languages, since BLIP2 and CLIP were trained on mostly English data. We believe that translating some synthetic captions to other languages to improve the representations of low-resource languages in the training set would be a valuable direction for future work. In general, the lack of standard multilingual tasks/ benchmarks has been a major challenge in evaluating CLIP’s capabilities in non-English settings.

---

> > ### Comment · Reviewer_fUUj · 2023-08-19
> >
> > Thank you for your comments. My score remains the same since my questions did not constitute a significant factor in a score change.

---

### Official Review · Reviewer_DaAH · 2023-07-21
**Investigating the tradeoff between using noisy webdata and synthetic captions is crucial. The study on mixing strategy is interesting.**

**Rating:** 6
**Confidence:** 4
**Correctness:** Yes, the claims made in the submissio…

**Strengths:**

The paper offers a comprehensive analysis of the task from four perspectives:

1. How to select a captioning model (based on which metric.)
2. How to developing a strategy to combine signals from multiple caption sources.
3. Investigating what makes synthetic captions effective.
4. Understanding how the benefits of synthetic captions scale up.

Through this analysis, several conclusions are reached:

1. Reference-free captioning metrics prove to be more reliable.
2. A hybrid strategy that combines raw and synthetic caption data can lead to sota performance.
3. Individual synthetic captions are typically less noisy and provide more visual information. However, at a population level, they tend to be less diverse than raw captions.
4. The quality control of images, as well as the diversity gap between model-generated and web-scraped captions, become increasingly crucial in large data scenarios.

The result is insightful for the future work on utilizing synthetic data, especially on synthetic captions for training VLMs.

**Additional Feedback:**

See “Opportunities For Improvement”

**Clarity:**

Yes, the idea is straightforward and the paper is easy to follow. There are many experiments to support the result.

**Documentation:**

There are some detail on data collection and organization (how to filter data). Documentation and intended uses, a URL for reviewer access to the dataset, and a hosting, licensing, and maintenance plan are not given.

**Limitations:**

See “Opportunities For Improvement”

**Opportunities For Improvement:**

1. Is the CIDEr score for OpenCLIP-CoCa, ViT-L/14 in Table 1 a typographical error (0.354)?

2. BLIP2 employs CLIP to rank generated captions, which could naturally cause the captions to be CLIP-preferred, as shown in Figure 4. Further investigations using other VLMs are necessary to consolidate this conclusion.

3. In Figure 7, authors claim, “Synthetic captions display a clear advantage over raw captions on retrieval tasks,” but a discussion on potential reasons for this observation is missing.

4. While zero-shot performance on ImageNet is a classic measure, it would be interesting to see if the conclusions could be extended to other datasets.

**Relation To Prior Work:**

Yes, prior works are discussed.

**Summary And Contributions:**

The paper first explored different mixing strategy for raw and synthetic captions (by using several image captioning models ) on different scales of the DataComp benchmark. Then, authors analyze what makes 12 synthetic captions effective. The importance of increasing data quantity is disscussed.

---

> ### Author Response · Authors · 2023-08-17
> **Response to Reviewer DaAH**
>
> We thank the reviewer for all the valuable feedback!
>
> **“Is the CIDEr score for OpenCLIP-CoCa, ViT-L/14 in Table 1 a typographical error (0.354)?”** We have confirmed this value with the author of OpenCLIP-CoCa. Since CIDEr seeks to measure the overlap between n-grams in the candidate captions and n-grams in the human-provided captions, a visual inspection of the 20 random examples we provided in Appendix A would show that the captions generated by the pre-trained OpenCLIP-CoCa model have very little n-gram overlap with the respective raw captions.
>
>
> **“BLIP2 employs CLIP to rank generated captions, which could naturally cause the captions to be CLIP-preferred, as shown in Figure 4. Further investigations using other VLMs are necessary to consolidate this conclusion.”** Thank you for raising this point. We agree that repeating the investigation with other VLMs would be useful in supporting this finding. Here we include *new results* showing the image-text cosine similarity distribution of captions generated by the GIT-large model [1] using the same generation hyperparameters as what we used for BLIP2 captions in our paper: https://postimg.cc/svyLRqwv. GIT-large was pre-trained on 20M image-text pairs and did not employ CLIP to select its pre-training data (refer to [1] for more details). We find that across 100K images we randomly selected, GIT-large captions still see higher image-text cosine similarity overall compared to the corresponding raw captions (mean 0.23 versus 0.20).
>
> [1] GIT: A Generative Image-to-text Transformer for Vision and Language. Wang et al., 2022.
>
> **“In Figure 7, authors claim, “Synthetic captions display a clear advantage over raw captions on retrieval tasks,” but a discussion on potential reasons for this observation is missing.”** We offer a few potential reasons here and will revise the paper accordingly: (i) higher image-text alignment in general helps boost retrieval capabilities, (ii) on a related note, synthetic captions are likely better at capturing prominent objects in the corresponding images than raw captions, which further helps with retrieval, (iii) captions from retrieval datasets are probably more similar to generated captions than typical web-crawled text.
>
>
> **“While zero-shot performance on ImageNet is a classic measure, it would be interesting to see if the conclusions could be extended to other datasets.”** We highlight ImageNet and retrieval tasks in our findings as these metrics may be of interest to many existing work. However, we have throughout the paper discussed “average performance” computed over 38 different tasks proposed by DataComp, as well as offered a fine-grained breakdown of performance impacts on individual tasks when BLIP2 captions are added to the training mix (see Figures 6 and 10). We note that the 38 tasks from the DataComp benchmark involve recognition and classification of a wide range of domains (e.g. texture/ traffic sign/ scene/ metastatic tissue/ geolocation/ animal/ etc.) in addition to image-text retrieval and commonsense association. The DataComp paper has also found that ImageNet accuracy is highly correlated with the average performance across all datasets.
>
> **Data release. NOTE: the dataset links in this response are not anonymized!**
> We have provided [a variety of BLIP2 captions](https://huggingface.co/datasets/thaottn/DataComp_medium_pool_BLIP2_captions) generated with 5 different softmax temperatures for the 128M images in the medium pool. We are in the process of releasing [1.28B BLIP2 captions](https://huggingface.co/datasets/thaottn/DataComp_large_pool_BLIP2_captions) corresponding to the images in the large pool of DataComp.

---

### Official Review · Reviewer_cRge · 2023-07-24
**Review of Improving multimodal datasets with image captioning**

**Rating:** 6
**Confidence:** 4

**Strengths:**

1. It serves as a first step towards improving the quality of web-scale datasets via the use of synthetic captions
2. By using generated captions, the authors are able to increase image-text relations while reducing the noise from low-quality captions.


**Additional Feedback:**

1. In section 5, do you maintain consistency in quantity on different ways of filtering and combining raw and generated captions to fairly compare?
2. Will the final mixed annotations or code be provided?


**Clarity:**

The paper is well-written overall, but there are still some issues that need to be addressed:
(1) Acronym usage: The paper should use the full name when using an acronym for the first time so that the readers can understand it. For example, NSFW (Not Safe For Work) in the introduction.
(2) Figure explanation missing: For some figures, it’s hard to understand only the figure without a full explanation. For example, in Fig.2, we cannot match the legend with your different ways of filtering and combining; what does the grounding ratio in Fig.3 mean?


**Correctness:**

The construction of the dataset is reasonable and considerate. The author tries to improve the quality of web-scale datasets via the use of synthetic captions.

**Documentation:**

The authors provide detailed details on producing the dataset.

**Ethics:**

I think that this manuscript does not have any ethical issues because this paper uses public datasets and benchmarks.

**Limitations:**

For different tasks or different data sets, the effectiveness of filtering methods is inconsistent, which needs to be further considered and resolved.

**Opportunities For Improvement:**

1. The role of the dataset has not been fully validated. This manuscript only verifies the image caption performance of the DataComp benchmark. I’ 'm glad to see it also valid the effectiveness of retrieval tasks. However, the effectiveness of this work may be validated by more models and more tasks.
2. The results of the manual verification mentioned in Fig.6 have not been fully validated. Although it can improve the 23 tasks in DataComp, it performs worse on the other 15 datasets, which is contrary to the previous statement. So, it should be discussed in more detail.


**Relation To Prior Work:**

Yes.

**Summary And Contributions:**

The research paper explores the use of generated captions to improve the quality of web-scraped image-text pairs. The authors experiment with different mixing strategies for raw and synthetic captions and achieve state-of-the-art performance on the DataComp benchmark, improving ImageNet accuracy and average accuracy over 38 tasks. They also find that the best-performing approach is 2x better at Flickr and MS-COCO retrieval tasks. The paper analyzes why synthetic captions are effective and explore the relationship between a model's performance on standard image captioning benchmarks and the usefulness of the captions it generates for multimodal training.

---

> ### Author Response · Authors · 2023-08-17
> **Response to Reviewer cRge**
>
> We would like to thank the reviewer for the comments and suggestions.
>
> **“The effectiveness of this work may be validated by more models and more tasks.”** The DataComp benchmark already contains 38 datasets that involve recognition and classification of a wide range of domains (e.g. texture/ traffic sign/ scene/ metastatic tissue/ geolocation/ animal/ etc.) in addition to image-text retrieval and commonsense association. Refer to Table 14 of the DataComp paper [1] for more details. Furthermore, [1] also shows that (i) ImageNet performance is highly correlated with the average performance across all 38 datasets (see Figure 24 of [1]), and (ii) the rankings of the baseline filtering strategies are relatively consistent across different architectures including ViT and ConvNeXt (Table 11 of [1]). This gives us confidence that the effectiveness of using synthetic captions will hold across a range of models and tasks.
>
> Per your comment, we include **new experiments with an additional task**, visual question answering, using the VQA v1 dataset (Antol et al., 2015). The best baseline using raw data from the medium scale, Raw (top 30% intersect IN1k), scores 0.0573 on this task, while our best baseline using both sources of captions, Raw (top 30%) + BLIP2 (70%, filtered), gets 0.0712. We repeat the analysis using the two models compared in Figure 6 of our paper that have similar ImageNet performance: one trained on only top 30% raw captions, and the other trained on only top 30% BLIP2 captions. The latter also outperforms the former on this task (0.0787 vs. 0.0652). Note that the limited performance of CLIP on VQA task is expected, as previous work [2,3] has shown that models trained on only image-text pairs often fail to learn sufficient reasoning skills to do well on visual question answering.
>
> [1] DataComp: In search of the next generation of multimodal datasets. Gadre et al., 2023.
>
> [2] How Much Can CLIP Benefit Vision-and-Language Tasks? Shen et al., 2021.
>
> [3] OBELISC: An Open Web-Scale Filtered Dataset of Interleaved Image-Text Documents. Laurencon et al., 2023.
>
> **“The results of the manual verification mentioned in Fig.6 have not been fully validated. Although it can improve the 23 tasks in DataComp, it performs worse on the other 15 datasets, which is contrary to the previous statement.”** We would appreciate more clarity on what the reviewer means by manual verification or what statement they are referring to by “contrary to the previous statement”. We want to clarify that in Lines 236-239 that reference Figure 6, we made no claim that adding BLIP2 captions helps on all tasks. The goal of this performance breakdown is to better understand on which other tasks BLIP2 captions could help significantly, by comparing a model trained on only raw captions and one trained on only BLIP2 captions, both having similar ImageNet accuracy.
>
> **“For different tasks or different data sets, the effectiveness of filtering methods is inconsistent, which needs to be further considered and resolved.”** We posit that it is unlikely that there exists a filtering strategy that yields the best performance on all tasks and datasets. In fact, a major contribution of our work is to demonstrate that by augmenting the existing candidate pool with synthetic captions, the best filtering and mixing strategy is different at different scales, suggesting that the importance of certain properties of the data distribution (e.g. having relevant captions or having object-centric images) varies depending on the training regime. We don’t view synthetic captions as a panacea for enhancing all image-text datasets. Instead, we believe there is still more room for improvement for synthetic captions to demonstrate comparable linguistic diversity as web-crawled captions at larger scales.
>
> **“The paper should use the full name when using an acronym for the first time so that the readers can understand it. For example, NSFW (Not Safe For Work).”** We thank the reviewer for the comment, and we will revise our manuscript accordingly.
>
> **“For some figures, it’s hard to understand only the figure without a full explanation. For example, in Fig.2, we cannot match the legend with your different ways of filtering and combining; what does the grounding ratio in Fig.3 mean?”** We have tried to summarize the experimental setup and key takeaways in the caption corresponding to each figure. Given space constraints, it is challenging to include sufficient explanation of various filtering methods for Figure 2. Regarding Figure 3, the grounding ratio has been explained in Line 208 of the main text. We allude to this in Figure 3’s caption by saying “the fraction of [words] being visual tokens in each caption”. We will expand on the legends to make them more self-contained in the next version of the paper.

---

> > ### Author Response · Authors · 2023-08-17
> > **Response to Reviewer cRge (cont.)**
> >
> > **“In section 5, do you maintain consistency in quantity on different ways of filtering and combining raw and generated captions to fairly compare?”** As stated in Section 3, the compute budget for each scale is fixed in order to isolate data quality as the main factor influencing performance. In other words, all data subsets resulting from filtering and combining caption sources may have different numbers of image-text pairs, but they are all trained for the same number of steps at each scale.
> >
> > **“Will the final mixed annotations or code be provided?” NOTE: the links we provide in this response are not anonymized!** Our experiments make use of open-source code: training code comes from DataComp and image captioning implementation comes from OpenCLIP and HuggingFace’s Transformer repositories. We provide the code for labeling the candidate pool with new caption metadata [here](https://github.com/mlfoundations/dataset2metadata/tree/main). We are in the process of releasing [1.28B BLIP2 captions](https://huggingface.co/datasets/thaottn/DataComp_large_pool_BLIP2_captions) corresponding to the images in the large pool of DataComp. We have in the meantime provided [a variety of BLIP2 captions](https://huggingface.co/datasets/thaottn/DataComp_medium_pool_BLIP2_captions) generated with 5 different softmax temperatures for the 128M images in the medium pool.

---

### Official Review · Reviewer_fLcW · 2023-07-28
**Synthetic image captioning gives new life to discarded images!**

**Rating:** 8
**Confidence:** 4
**Clarity:** The paper is very clear.

**Strengths:**

This paper has several strengths:
- This is a well written and clearly explained paper in all aspects
- The work is a valuable contribution to the field of dataset creation
- The ablations, analyses, and experiments are performed soundly and are of high quality.
- Achieving SOTA performance on their chosen benchmarks proves the papers utility.

**Additional Feedback:**

N/A

**Correctness:**

The claims made by the work appear correct, and the construction is sound alongside the experimental setup.

**Documentation:**

There is reasonable documentation for the dataset/project as a whole. However, increased visibility into the dataset (like a viewer) or releasing the tooling to expand other datasets would be a valuable addition.

**Ethics:**

The paper does not contain significant ethical concerns.

**Limitations:**

The authors discuss the limitations of their work in a sound manner.

**Opportunities For Improvement:**

I would like to see the code for this project publically released and easily usable by other research teams who would like to employ the methods in this paper to expand their datasets.

**Relation To Prior Work:**

The paper cites prior work sufficiently. There is lots of reference to previous work that is similar.

**Summary And Contributions:**

This paper is incredibly high quality and offers a very strong piece of utility to the community. The method of using image captioning models to give new use for images with poor caption quality clearly results in increased performance and provides a useful new method for other datasets to employ.

---

> ### Author Response · Authors · 2023-08-17
> **Response to Reviewer fLcW**
>
> Thank you for your time and feedback! We appreciate your recognition of the strengths of the paper.
>
> **Code and data release. NOTE: the links we provide in this response are not anonymized!**
>
> Our experiments make use of open-source code: training code comes from DataComp and caption generation comes from OpenCLIP and HuggingFace’s Transformer repositories. We provide the code for labeling the candidate pool with new caption metadata [here](https://github.com/mlfoundations/dataset2metadata/tree/main).
>
> We have provided a variety of [BLIP2 captions](https://huggingface.co/datasets/thaottn/DataComp_medium_pool_BLIP2_captions) generated with 5 different softmax temperatures for the 128M images in the medium pool. We are in the process of releasing [1.28B BLIP2 captions](https://huggingface.co/datasets/thaottn/DataComp_large_pool_BLIP2_captions) corresponding to the images in the large pool of DataComp.

---

### Author Response · Authors · 2023-08-17
**Summary of new results and data release**

We would like to thank all reviewers again for providing thoughtful reviews of our work. Here we highlight several new results/ changes taking your feedback into consideration:
- Besides the 38 classification and retrieval tasks proposed by the DataComp benchmark, we have also looked into an additional task, visual question answering, using the VQA v1 dataset (Antol et al., 2015). We find that using BLIP2 captions in the training data consistently yields improvement on this task compared to using only raw captions. Specifically, at medium scale, our best baseline using both sources of captions, Raw (top 30%) + BLIP2 (70%, filtered), gets 0.0712, while the best baseline using raw captions, Raw (top 30% intersect IN1k), scores 0.0573, and another competitive baseline, Raw (top 30%), scores 0.0652.
- Given a reviewer’s concern about potential biases from using synthetic captions, we break down the performance of some models trained on either or both sources of captions, for predicting the race and gender annotated in the Fairface dataset (Karkkainen et al., 2021). We find that using synthetic captions improves the performance of the disadvantaged group (e.g. female) significantly, and reduces the performance gap between male and female groups while still boosting the overall performance on all race categories. Extending this analysis to other kinds of biases and fairness datasets is a highly important direction for future work.
|Gender|Model|Race| | | | | | |
|:----|:----|:----|:----|:----|:----|:----|:----|:----|
| | |Black|White|Indian|Latino/ Hispanic|Middle Eastern|South East Asian|East Asian|
|Male|Raw (top 30%)|93.0|88.8|91.2|90.8|92.3|85.3|81.3|
| |BLIP2 (top 30%)|87.2|73.7|77.2|74.9|78.6|72.0|64.0|
| |Raw (top 30%) + BLIP2 (70%, filtered)|90.5|75|79.7|79.4|81.1|72.4|65.3|
|Female|Raw (top 30%)|20.3|47.1|35.1|42.0|40.9|44.9|56.8|
| |BLIP2 (top 30%)|36.9|70.8|57.9|67.5|67.4|64.1|78.4|
| |Raw (top 30%) + BLIP2 (70%, filtered)|32.9|74.8|56.5|66.3|67.9|67.8|81.9|
|Overall|Raw (top 30%)|56.7|68|63.2|66.4|66.6|65.1|69.1|
| |BLIP2 (top 30%)|62.1|72.3|67.6|71.2|73.0|68.1|71.2|
| |Raw (top 30%) + BLIP2 (70%, filtered)|61.7|74.9|68.1|72.9|74.5|70.1|73.6|
- Last but not least, we have done code and data release. **Note that the links provided are not anonymized**. We provide the code to label images in web datasets with BLIP2 captions [here](https://github.com/mlfoundations/dataset2metadata/tree/main). We are also releasing corresponding BLIP2 captions for the [medium (128M)](https://huggingface.co/datasets/thaottn/DataComp_medium_pool_BLIP2_captions) and [large (1.28B)](https://huggingface.co/datasets/thaottn/DataComp_large_pool_BLIP2_captions) pools of DataComp (the former pool comes with 5 synthetic captions per image, given our experiments with using different softmax temperatures for caption generation).

---

### Decision · Program_Chairs · 2023-09-22

**Decision:**

Accept (Poster)

**Comment:**

The paper has received a consistent set of review comments, including two strong acceptance reviews and three weak acceptance review. most reviewers have praised the substantial contributions made in terms of data collection, ablations, analysis and experiments. In light of these positive assessments, I believe the paper merits an  'accept' recommendation.